# Selective usage of ANP32 proteins by influenza B virus polymerase: Implications in determination of host range

Zhenyu Zhang[1][☯], Haili Zhang[1][☯], Ling Xu[1], Xing Guo[1], Wenfei Wang[2], Yujie Ji[1], Chaohui Lin[1], Yujie Wang[1], Xiaojun Wang[1]*

1 State Key Laboratory of Veterinary Biotechnology, Harbin Veterinary Research Institute, the Chinese Academy of Agricultural Sciences, Harbin, P. R. China, 2 School of Life Science, Northeast Agricultural University, Harbin, P. R. China

☯ These authors contributed equally to this work.
* wangxiaojun@caas.cn

**Data Availability Statement:** All relevant data are within the manuscript and its Supporting Information files.

## Abstract

The influenza B virus (IBV) causes seasonal influenza and has accounted for an increasing proportion of influenza outbreaks. IBV mainly causes human infections and has not been found to spread in poultry. The replication mechanism and the determinants of interspecies transmission of IBV are largely unknown. In this study, we found that the host ANP32 proteins are required for the function of the IBV polymerase. Human ANP32A/B strongly supports IBV replication, while ANP32E has a limited role. Unlike human ANP32A/B, chicken ANP32A has low support activity to IBV polymerase because of a unique 33-amino-acid insert, which, in contrast, exhibits species specific support to avian influenza A virus (IAV) replication. Chicken ANP32B and ANP32E have even lower activity compared with human ANP32B/E due to specific amino acid substitutions at sites 129–130. We further revealed that the sites 129–130 affect the binding ability of ANP32B/E to IBV polymerase, while the 33-amino-acid insert of chicken ANP32A reduces its binding stability and affinity. Taken together, the features of avian ANP32 proteins limited their abilities to support IBV polymerase, which could prevent efficient replication of IBV in chicken cells. Our results illustrate roles of ANP32 proteins in supporting IBV replication and may help to understand the ineffective replication of IBV in birds.

## Author summary

Influenza B viruses infect humans and few other mammals, but fairly rare in birds. Here we found that IBV requires the involvement of host ANP32 proteins in the replication process, in which ANP32A and ANP32B play major roles and can fully support polymerase activity independently, while ANP32E gives only limited support to IBV polymerase because of certain substitutions compared with ANP32A/B. Chicken ANP32A has a 33-amino-acid insert not present in mammals and provides better support to avian IAV polymerase, but this insert impairs its support for IBV polymerase activity. Chicken ANP32B

**Funding:** This work was supported by the grants from the Natural Science Foundation of China (www.nsfc.gov.cn) to X.W. (as a part of grant to Dr. Hualan Chen's grant: 31521005) and Z.Z. (31702269) and the Natural Science Foundation of Heilongjiang Province (http://jj.hljkj.cn/zr/) to X.W. (No. JC2018010).The funders had no role in study design, data collection and analysis, decision to publish, or preparation of the manuscript.

**Competing interests:** The authors have declared that no competing interests exist.

and ANP32E have even lower support to IBV polymerase due to specific inactive mutations at sites 129/130. Our findings reveal an important role for ANP32 proteins in IBV polymerase activity and suggest the possible molecular basis of adaptation and restriction of IBV infection in different species.

## Introduction

Influenza A and Influenza B are major infectious respiratory tract diseases and cause significant morbidity and mortality in humans. The influenza A virus (IAV) and influenza B virus (IBV) belong to the group orthomyxoviridae, which are characterized by a single-stranded segmented RNA genome and enveloped spherical or filamentous particles with studded surface proteins [1, 2]. Influenza A viruses can infect a broad range of hosts including humans, other mammals, and birds. Based on antigenicity of the surface glycoproteins hemagglutinin (HA) and neuraminidase (NA), 18 HA subtypes (H1-H18) and 11 NA subtypes (N1-N11) have been discovered on the influenza A viruses isolated from different hosts. Unlike influenza A viruses, all influenza B viruses have been identified as belonging to two genetic and antigenic lineages (based on the HA protein): the B/Yamagata lineage and the B/Victoria lineage, and both lineages circulate mainly in the human population [3]. IAVs of subtypes H1N1 and H3N2, and IBVs of both lineages are the main pathogens responsible for seasonal influenza [4]. It is estimated that in the US 2019–2020 season, there were at least 36 million cases of illness and 22,000 deaths from flu, of which half were caused by IBV (https://www.cdc.gov/flu/weekly/index.htm#ILIMap).

Mammalian cells have been commonly observed to have a restriction effect on infection by avian IAV and the mechanism of this interspecific infection restriction of IAV has been long studied. Some H1, H5 or H7 subtype IAV viruses can overcome the restriction by evolving certain mutations of polymerase, such as PB2 E627K or G590S and Q591R, and gain ability to replicate in mammals [5–10]. One of the main barriers to avian IAV infection of mammals is that avian viral polymerase is poorly adapted to the host acidic nuclear phosphoprotein 32 family member A (ANP32A) molecules [11–17]. Most avian ANP32A has a special 33-amino-acid insert which enhances its ability to support polymerase of IAVs from both mammals and birds. Mammalian ANP32A and ANP32B without the 33-amino-acid insert do not support avian viral polymerase [11]. The ANP32 family includes three conserved family members: ANP32A, ANP32B, and ANP32E. Previously we and other lab have identified ANP32A and ANP32B as the host molecules critical for determining the polymerase activity of influenza A viruses in different hosts [11, 13, 14], and among mammalian ANP32 proteins the swine ANP32A shows unique feather in supporting chicken AIV replication in pig cells [18]. Three splice variants of ANP32A in avian species that harbor 33, 29, or no special amino acid insertion have been identified and showed different supports to avian polymerase. In some avian species like swallow and goose the shorter variants may help to drive or maintain some mammalian-adaptive IAV polymerase mutations [12, 15].

IBVs are believed to be stably adapted to humans and are continually circulating in the human population. IBVs have also occasionally been isolated from other mammals, including dogs, pigs, and harbor seals [19–22], and there are reports from serological evidence of influenza B infection in dogs, guinea pigs, ruminants, and chimpanzees [23–26]. One early paper reported that zoo birds were infected with influenza B viruses [27], but there have been no similar reports since 1980s. This suggests that mammals are more susceptible to IBV than birds, but the mechanism or the reasons for this have never been clear.

Here we created different ANP32 protein knockout cell lines, and demonstrated that human ANP32A and ANP32B proteins are required for the polymerase activity of the influenza B virus. Unlike the polymerase of IAV, the IBV polymerase can use ANP32E, albeit with limited activity. Intriguingly, we found that chicken ANP32A has a much lower ability to support IBV polymerase activity than does mammalian ANP32A, mainly because it harbors a 33-amino-acid insert not present in mammals, which is required to support avian IAV replication. These 33 amino acids give an advantage to avian IAV, but in IBV, they become a barrier. We also found that although chicken ANP32B and ANP32E are all in "short-form" like those proteins from humans, they both have functionally inactive mutations at sites 129/130, resulting in the loss of support to the IBV polymerase. These findings reveal an important role for ANP32 proteins in IBV polymerase activity and suggest the molecular basis of restriction of IBV infection in chickens.

## Results

### Human ANP32A, ANP32B, and ANP32E support influenza B viral polymerase activity to different degrees

IBV has similar replication mechanism to that of IAV. Our previous work and results from other labs showed that ANP32A&B proteins play a crucial role in supporting the polymerase activity of IAV [14, 28], and we wanted to investigate whether ANP32 proteins were also indispensable for IBV replication. We had previously constructed 293T knockout cell lines, including ANP32A knockout (AKO), ANP32B knockout (BKO), and ANP32A and ANP32B double knockout (DKO) cell lines, and identified that the polymerases from human seasonal H1N1 influenza viruses and human adapted H7N9 virus have similar dependence on the ANP32 proteins [14]. Here we confirmed that the polymerase of IAV H7N9$_{AH13}$ had similar levels of activity in AKO cells, BKO cells, and wild type 293T cells. However, in DKO cells, the H7N9$_{AH13}$ viral polymerase activity decreased sharply by about 5000-fold, which was comparable to the background value of polymerase activity (293T del PB2) (Fig 1A). The activity of the IBV polymerase reduced by 100-fold in DKO cells, but was still much higher than the background value (293T del PB2) (Fig 1A). We speculated therefore that the conserved ANP32 family member, ANP32E, may contribute to the support of IBV polymerase activity. ANP32E has a similar sequence to those of both ANP32A and ANP32B, but is not able to support IAV replication [14, 29]. Therefore, based on our DKO cell line, we knocked out ANP32E to make an ANP32A, ANP32B, and ANP32E triple knockout (TKO) cell line (Fig 1B). The viability of TKO cells is as good as the 293T cells (S1 Fig). We found that IAV polymerase activity showed no significant differences between DKO and TKO cells, while the polymerase activity of IBV decreased significantly (10-fold) in TKO cells compared with that in DKO cells (Fig 1C). These results suggested that ANP32E is able to give certain support for IBV polymerase activity, although the support is mild. To investigate the contribution of ANP32 proteins in supporting viral replication, an IBV strain B/Yamagata/PJ/2018, which showed comparable replication efficiency in 293T cells with that in MDCK cells (S2 Fig), was used to infect 293T cells and the ANP32 protein knock out cells. We found that B/Yamagata/PJ/2018 could replicate well in wild-type 293T, AKO, and BKO cells, but not in DKO or TKO cells (Fig 1D). Reconstitution of either ANP32A or ANP32B in TKO cells completely restored the IBV polymerase activity, but reconstitution of ANP32E was able only partially to restore the viral polymerase activity (Fig 1E). Dose-dependent experiments showed that transfection of 10 nanograms of pCAGGS-ANP32A or ANP32B was enough to recover IBV polymerase activity (S3A and S3B Fig), while even high doses of ANP32E were not able to restore polymerase activity completely (S3C Fig). These results confirm that ANP32A and ANP32B play a major

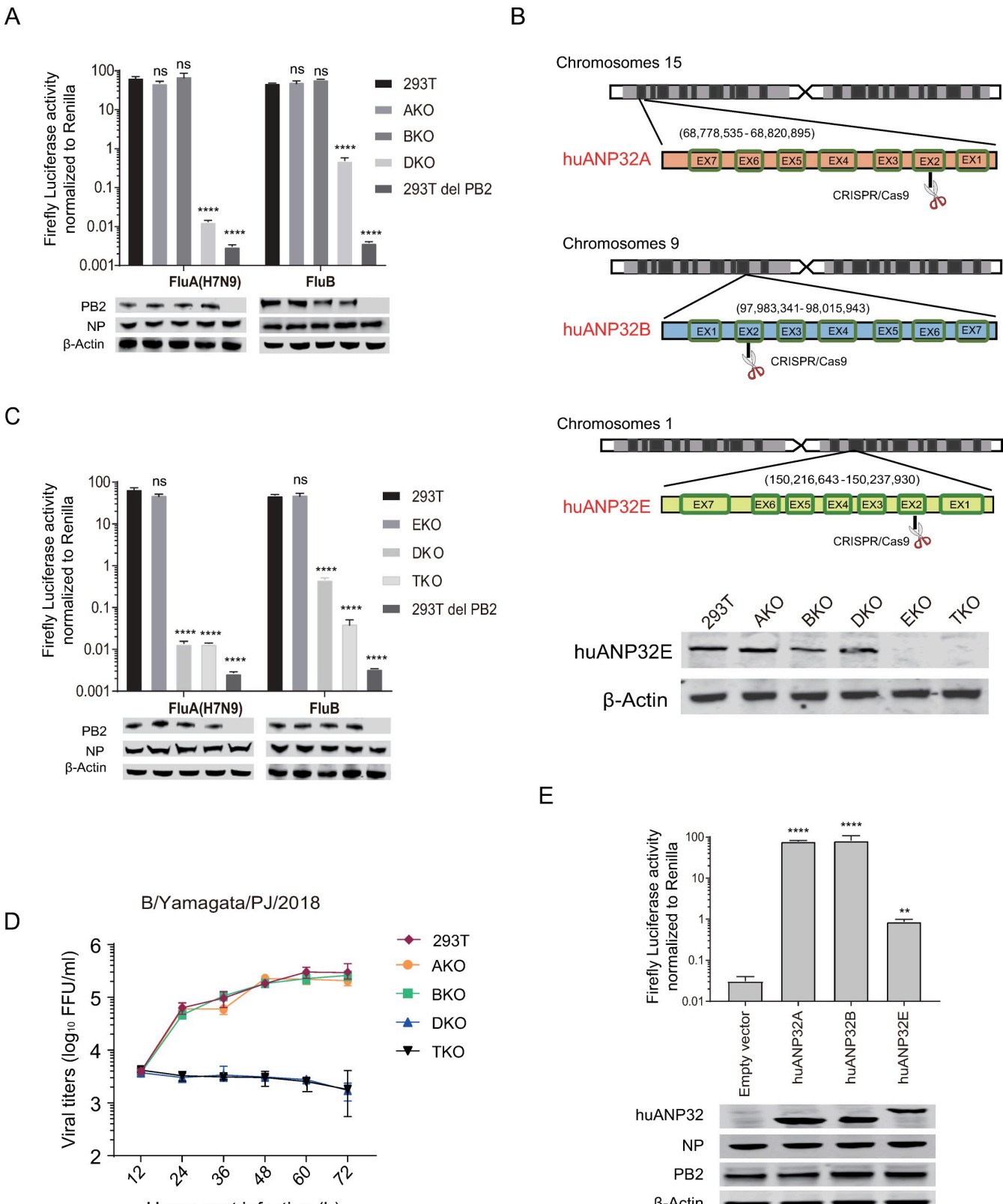

**Fig 1. ANP32A, ANP32B, and ANP32E support influenza B viral polymerase activity to different degrees. (A)** 293T, huANP32A knockout cells (AKO), huANP32B knockout cells (BKO), and huANP32A&B double knockout cells (DKO) were transfected with firefly minigenome reporter, *Renilla* expression control, and B/Yamagata/1/73 or H7N9$_{AH13}$ polymerase. As a negative control, 293T cells were transfected with the same plasmids, with the exception of the

PB2 expression plasmid. (**B**) Schematic diagram of gene analysis of human ANP32A, ANP32B and ANP32E sgRNA target positions in the chromosomes. (**C**) 293T, huANP32E knockout cells (EKO), huANP32A&B double knockout cells (DKO), and huANP32A&B&E triple knockout cells (TKO) were transfected with firefly minigenome reporter, *Renilla* expression control, and B/Yamagata/1/73 or H7N9$_{AH13}$ polymerase. As a negative control, 293T cells were transfected with the same plasmids, with the exception of the PB2 expression plasmid. (**D**) 293T, AKO, BKO, DKO, and TKO cells were infected with B/Yamagata/1/73 virus at a MOI of 0.1. The supernatants were sampled at 12, 24, 36, 48, 60, 72 h post infection and the viral titers were determined using Fluorescence Focus Units (FFU) assay on MDCK cells. The result is shown as average of n = 3 ± SD. (**E**) TKO cells were co-transfected with B/Yamagata/1/73 polymerase, minigenome reporter, and *Renilla* expression control together with 10 ng huANP32A, 10 ng huANP32B, 10 ng huANP32E, or 10 ng empty vector, and luciferase activity was assayed at 24 h after transfection. The expression of ANP32 proteins and polymerase was assessed using western blotting. The data indicate the firefly activity normalized to *Renilla*, Statistical differences between cells are labeled according to a one-way ANOVA followed by a Dunnett's test (NS = not significant, $^{**}P < 0.01$, $^{***}P < 0.001$, $^{****}P < 0.0001$). Error bars represent the SD of the replicates within one representative experiment.

role in the support of IBV replication through promoting polymerase activity; however, ANP32E shows limited IBV polymerase support, which is not enough to support IBV replication at its natural expression level in DKO cells.

## Species-specific support of influenza B viral polymerase activity by ANP32A proteins from different animals

IBV mainly infects humans and a few other mammals, and there is no strong evidence of infection in birds, indicating a species specific evolution pattern of IBV. Recently, ANP32 proteins have been reported as key host factors limiting the spread of IAV from birds to mammals. Whether ANP32 proteins from different species give differing supports to IBV, and whether they can act as a barrier limiting the spread of IBV between species, are unclear. We compared the ability of human and chicken ANP32 proteins to support IBV polymerase activity in TKO cells, and we found that human ANP32A (huANP32A) and ANP32B (huANP32B) gave strong support to IBV polymerase, but chicken ANP32A (chANP32A), chicken ANP32B (chANP32B), and chicken ANP32E (chANP32E) gave only weak support to IBV replication, levels which were comparable to human ANP32E (huANP32E) (Fig 2A). Similarly, in wild-type chicken DF-1 cells, overexpression of huANP32A or huANP32B dramatically increased IBV polymerase activity, while chANP32A and huANP32E exhibited weak support. Surprisingly, the chANP32B or chANP32E had a negative effect on IBV polymerase activity, which could be a negative effect caused by over expression of a non-functional ANP32 protein (Fig 2B). Furthermore, we observed overexpression of huANP32B in DF1 cells could enhance IBV infectivity (Fig 2C). The above results confirmed that huANP32A&B give strong but species specific support to polymerase of IBV compared with the chicken proteins chANP32A&B.

Because ANP32A proteins from humans and other mammals are in "short form" compared with chANP32A, we next investigated the support given by ANP32A proteins from different species to the polymerase activity of the two IBV strains B/Yamagata/1/73 and B/Victoria/Brisbane/60/2008. We found that ANP32A from mammals (human, pig, horse, and dog) and ostrich, that are all in "short form" without 33-amino-acid insert (S4 Fig), supported IBV polymerase activity significantly more than the ANP32As from poultry (duck, turkey, and chicken) or finch, which all have a 33-amino-acid insert (Fig 3A and 3B). Sequence alignment of mammalian with avian revealed that avian ANP32A contains an additional 33-amino-acid insert comprising a predicted SUMO interaction motif-like sequence (SIM) and a 27 amino acid repeat sequence [12, 13] (Fig 3C). We next investigated whether the 33 additional amino acids were responsible for the difference in activity between the ANP32A proteins. Insertion of the avian-specific 33 amino acids into human ANP32A (huANP32A$_{+33}$) reduced the polymerase activity in TKO cells, giving it an activity similar to chicken ANP32A. Conversely, deletion of the 33 amino acids from chicken ANP32A (chANP32A$_{\Delta 33}$) increased the polymerase activity to a level similar to that of huANP32A (Fig 3D and 3E). This suggests that the 33 amino acids

A

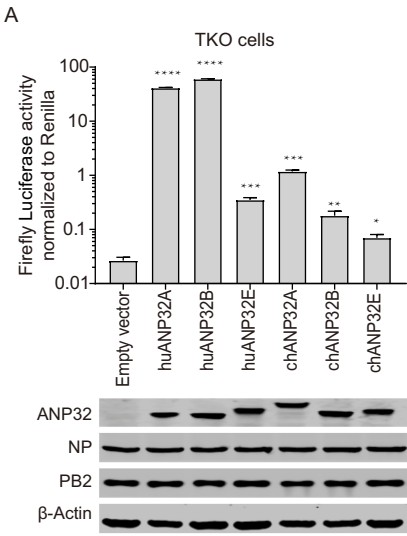

B

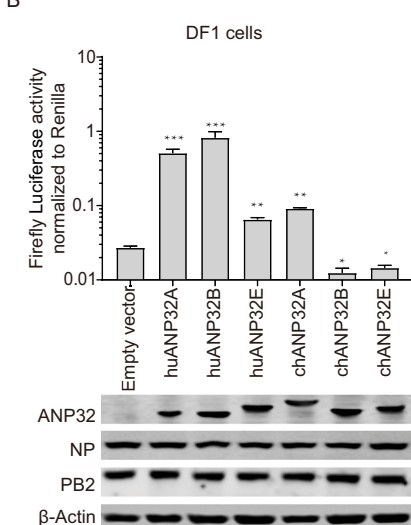

C

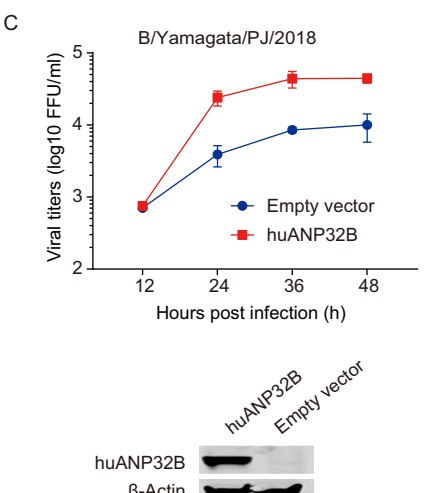

**Fig 2. Species-specific support of influenza B viral polymerase activity by ANP32 proteins from different animals.** (**A**) TKO cells were co-transfected with B/Yamagata/1/73 polymerase, minigenome reporter, *Renilla* expression control and 10 ng of one of the following: huANP32A, huANP32B, huANP32E, chANP32A, chANP32B, chANP32E or 10 ng empty vector. Luciferase activity was assayed at 24 h after transfection. The expression of ANP32 proteins and polymerase was assessed using western blotting. (**B**) DF1 cells were co-transfected with B/Yamagata/1/73 polymerase, minigenome reporter with chicken polI promoter, *Renilla* expression control and 10 ng of one of the following: huANP32A-flag, huANP32B-flag, huANP32E-flag, chANP32A-flag, chANP32B-flag, chANP32E-flag or 10 ng empty vectors. Luciferase activity was assayed at 24 h after transfection. The protein expression was determined by western blotting using different antibodies: anti-flag antibody for ANP32 proteins, and specific antibodies to polymerase and β-actin. The data indicate the firefly activity normalized to *Renilla*, Statistical differences between cells are labeled according to a one-way ANOVA followed by a Dunnett's test (NS = not significant, **$P < 0.01$, ***$P < 0.001$, ****$P < 0.0001$). Error bars represent the SD of the replicates within one representative experiment. (**C**) DF1 cells were transfected with 1 μg huANP32B-flag or empty vector in 6 well plate. Twenty-four hours post transfection DF1 cells were infected with B/Yamagata/PJ/2018 virus at a MOI of 0.1 and cultured at 33°C or 37°C. The supernatants were sampled at 12, 24, 36, 48 h post infection and the viral titers were determined using Fluorescence Focus Units (FFU) assay on MDCK cells. The expression of huANP32B was assessed by western blotting using anti-flag antibody. The result is shown as average of n = 3 ± SD.

difference between mammals and most birds is indeed an important domain in determining the different activities of IBV polymerase. Interestingly, this phenomenon reversed in IAV, where avian ANP32A proteins with the 33-amino-acid insert support the replication of avian-sourced IAV as well as human strains.

The 33 extra amino acids comprise 27 amino acids identical to those in the section neighboring the insert, together with 6 specific amino acids. Previous research has shown that 4 of these 6 amino acids (VLSL) make up a SUMO-interaction motif, and substitution or deletion of these four amino acids can weaken the replication of the avian influenza virus [12, 13]. Chickens have an ANP32A isoform that lacks these four hydrophobic residues, which slightly reduces its support to avian IAV polymerase [12, 13]. We found that deletion of this SUMO interaction motif (SIM)-like sequence from chANP32A (chANP32AΔSIM) could not alter IBV polymerase activity (Fig 3F). In conclusion, the ability of chANP32A to support IBV polymerase activity is much lower than that of mammalian ANP32A, because of a 33 or 29 amino acid insert in chANP32A. This result provides evidence of differential usage of ANP32A by IAV and IBV and indicates a potential role of the 33-amino-acid insert in species specific host-virus selection and evolution pattern.

## Avian ANP32B has low support of influenza B viral polymerase

All ANP32B proteins from different mammalian species lack the 33-amino-acid insert compared with chANP32A, although the chANP32B has a longer C-terminal of LCAR region than that of huANP32B. We showed that huANP32B gives strong support to IBV replication, while chANP32B gives only limited support to the IBV polymerase (Fig 4A and 4B). Our previous studies have demonstrated that chANP32B is naturally non-functional and cannot support the activity of IAV polymerase at all, because it lacks the 129N/130D functional signature found in other ANP32Bs, having instead 129I/130N, but not the longer LCAR tail [14]. Murine ANP32B (muANP32B), which encodes 129S/130D different from the more common 129N/130D found in huANP32B, huANP32A, and chANP32A, had been reported to support both IAV and IBV polymerase [28], could support IBV polymerase at similar level as huANP32B (Fig 4A and 4B). In order to verify whether this phenomenon was caused by the difference in 129/130 sites, huANP32B_N129I/D130N and chANP32B_I129N/N130D mutants were constructed and tested. The results show that the activity of the viral polymerase supported by huANP32B_N129I/D130N is significantly decreased compared with huANP32B, to the same extent as chANP32B. However, chANP32B I129N/N130D had a significantly increased viral polymerase activity compared with chANP32B, which was similar to the activity of huANP32B

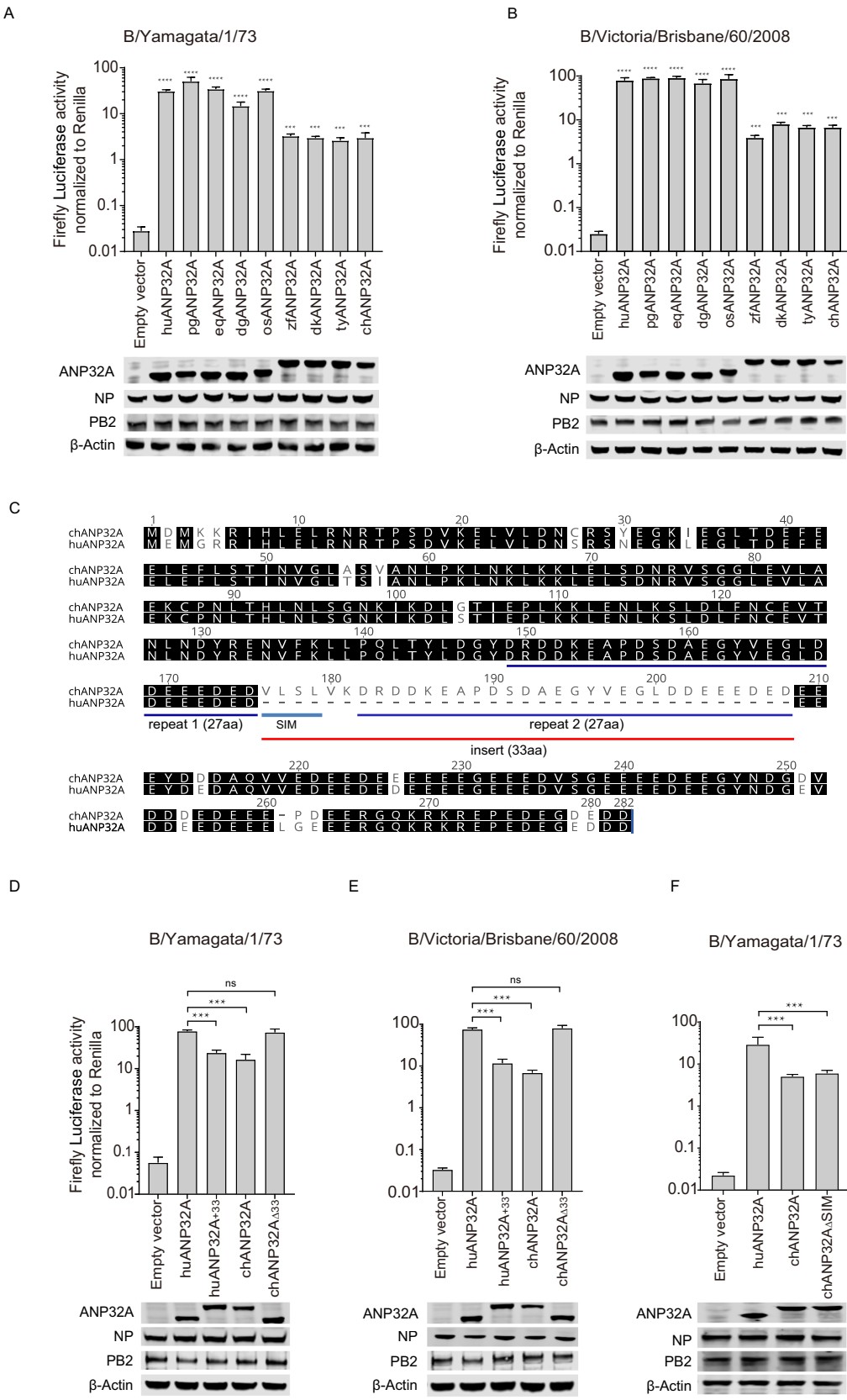

**Fig 3. Support of influenza B viral replication by ANP32A from different species and the key amino acids responsible for the support.** (**A and B**) TKO cells were co-transfected with 10 ng of ANP32A from different species or empty vector with minigenome reporter, *Renilla* expression control, influenza B virus polymerase from B/Yamagata/1/73 (**A**) or B/Victoria/Brisbane/60/2008 (**B**). (**C**) ANP32A amino acid sequences from humans and chicken were aligned using the Geneious R10 software. Gaps are marked with dashes. (**D to F**) TKO cells were co-transfected with 10 ng of huANP32A or chANP32A or the indicated mutations with minigenome reporter, *Renilla* expression control, and influenza B virus polymerase of either B/Yamagata/1/73 (**D and F**), or B/Victoria/Brisbane/60/2008 (**E**). The expression of ANP32 proteins and polymerase was assessed using western blotting. Luciferase activity was measured 24 h following transfection. The data indicate the firefly activity normalized to *Renilla*, Statistical differences between cells are labeled according to a one-way ANOVA followed by a Dunnett's test (NS = not significant, **P < 0.01, ***P < 0.001, ****P < 0.0001). Error bars represent the SD of the replicates within one representative experiment. pg, pig; eq, equine; dg, dog; os, ostrich; zf, zebra finch; dk, duck; ty, turkey; huANP32A$_{+33}$, huANP32A with the 33-amino-acid insert from chANP32A; chANP32A$_{\Delta 33}$, chANP32A without the 33-amino-acid insert missing in huANP32A; chANP32A$_{\Delta SIM}$, chANP32A without the SIM(VLSL) sequence which is missing in huANP32A.

(Fig 4C and 4D). These results show that chicken ANP32B has limited support for IBV polymerase activity compared with that from mammals, and that this effect is caused by the different amino acids at positions 129/130.

## Avian ANP32E has a limited ability to support influenza B viral polymerase activity

HuANP32E is an important member of the ANP32 family and has been shown to have histone chaperone activity [30, 31]. We found that huANP32E has no impact on IAV polymerase activity but gives limited support to IBV polymerase activity (Fig 1 and Fig 2). However, whether ANP32E proteins from other animals, especially those from birds, can support IBV replication is largely unknown. Alignment of ANP32E sequences from different species indicated that there were two major differences in the C-terminus of ANP32E between mammals and birds. The first one was located upstream of the 200$^{th}$ amino acid, presenting a consecutive ten acidic amino acids deletion in birds; the second one was located downstream of the 200$^{th}$ amino acid, with eight consecutive amino acids differences between mammals and birds (Fig 5A). We first compared the support of ANP32Es from various species for IBV polymerase activity. The results showed that the ability of ANP32Es from mammals (human, pig, horse, dog, and mouse) to support IBV polymerase activity was nearly 10 times higher than that of ANP32Es from birds (zebra finch, duck, turkey, and chicken) (Fig 5B and 5C). To map the critical residues that determine the differences between mammalian and avian ANP32Es, we generated and tested certain chimeric clones between huANP32E and chANP32E (Fig 5D). We found that replacement of the 140 C-terminal amino acids of huANP32E with those of chANP32E (hu-chANP32E) reduces the activity of huANP32E to a level to that of chANP32E, and conversely, replacement of the C-terminus of chANP32E with that of huANP32E (ch-huANP32E) increases the level of activity chANP32E to that of huANP32E (Fig 5E). Exchanging of the N-terminal 200 amino acids between chANP32E and huANP32E revealed that the key region determining the differences in ability to support IBV polymerase activity was located between the 140$^{th}$ and the 200$^{th}$ amino acid, in the consecutive acidic amino acid (D & E amino acid) insertion region. To verify this, we conducted our transfection experiments using four mutants: 10-amino-acid deletion in huANP32E (huANP32E_Δ10aa), 10-amino-acid insert in chANP32E (chANP32E_10aa+), and an 8-amino-acid replacement between huANP32E and chANP32E (huANP32E_8aamut and chANP32E_8aamut). The results showed that huANP32E_Δ10aa had similar ability to support IBV polymerase to that of chANP32E; chANP32E_10aa+ had similar activity to huANP32E; and that the 8-amino-acid replacement did not change the activity of huANP32E and chANP32E at all (Fig 5E). These results suggested that the deletion of the consecutive acidic amino acid (D & E amino acid)

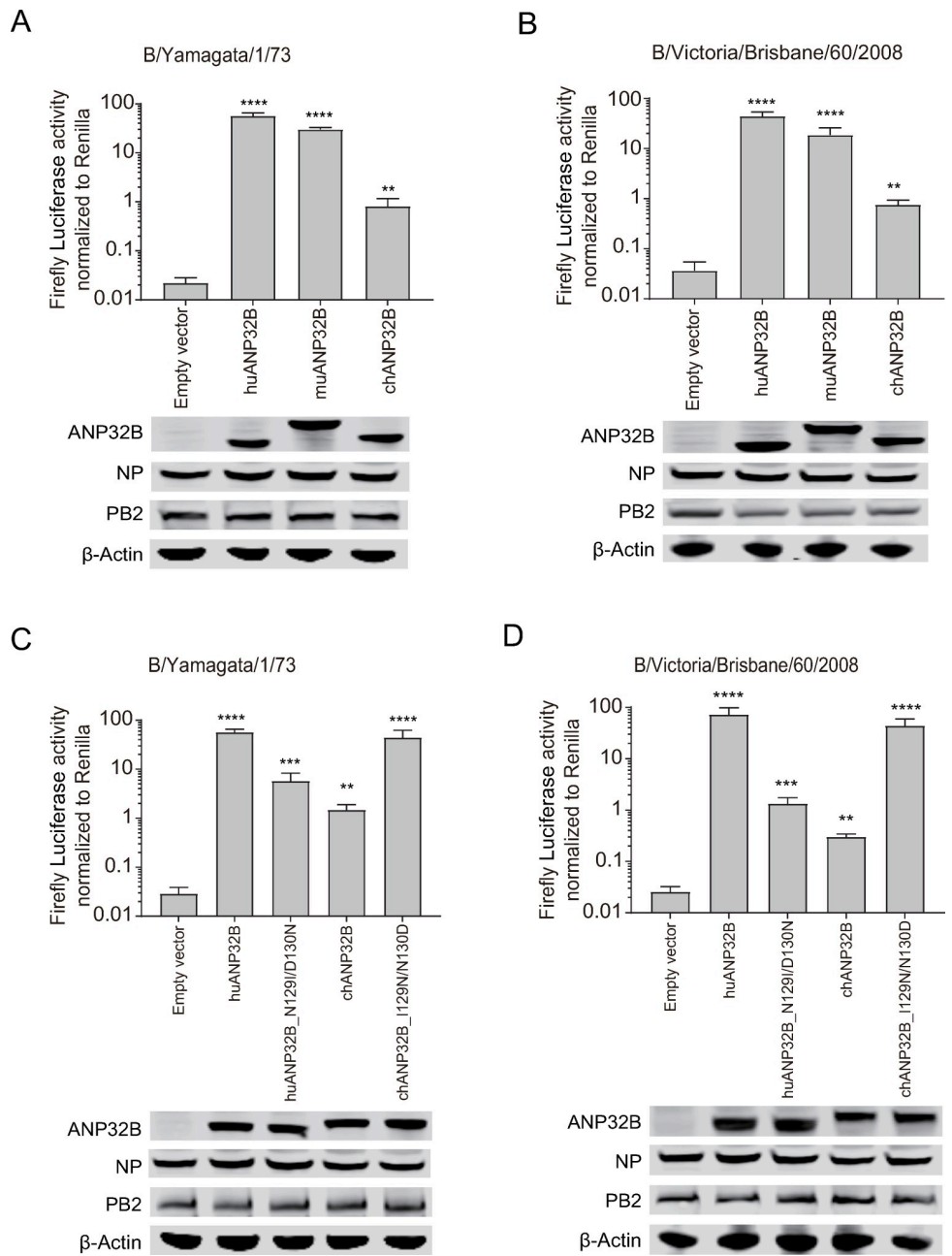

**Fig 4. The 129/130 site of ANP32B determines the supporting of influenza B viral polymerase.** TKO cells were co-transfected with 10 ng of either ANP32B from different species or empty vector was co-transfected with minigenome reporter, *Renilla* expression control, and influenza B virus polymerase from B/Yamagata/1/73 (**A**), or B/Victoria/Brisbane/60/2008 (**B**). HuANP32B or chANP32B or the indicated mutations was co-transfected with minigenome reporter, *Renilla* expression control, influenza B virus polymerase of B/Yamagata/1/73 (**C**), B/Victoria/Brisbane/60/2008 (**D**). The expression of ANP32 proteins and polymerase was assessed using western blotting. Luciferase activity was measured 24 h after transfection. The data indicate the firefly activity normalized to *Renilla*, Statistical differences between cells are labeled according to a one-way ANOVA followed by a Dunnett's test (NS = not significant, **P < 0.01, ***P < 0.001, ****P < 0.0001). Error bars represent the SD of the replicates within one representative experiment.

region in chANP32E was the main cause for the lower activity. We also noticed murine ANP32E (muANP32E) lacks the 10-amino-acid insert, which is different from other

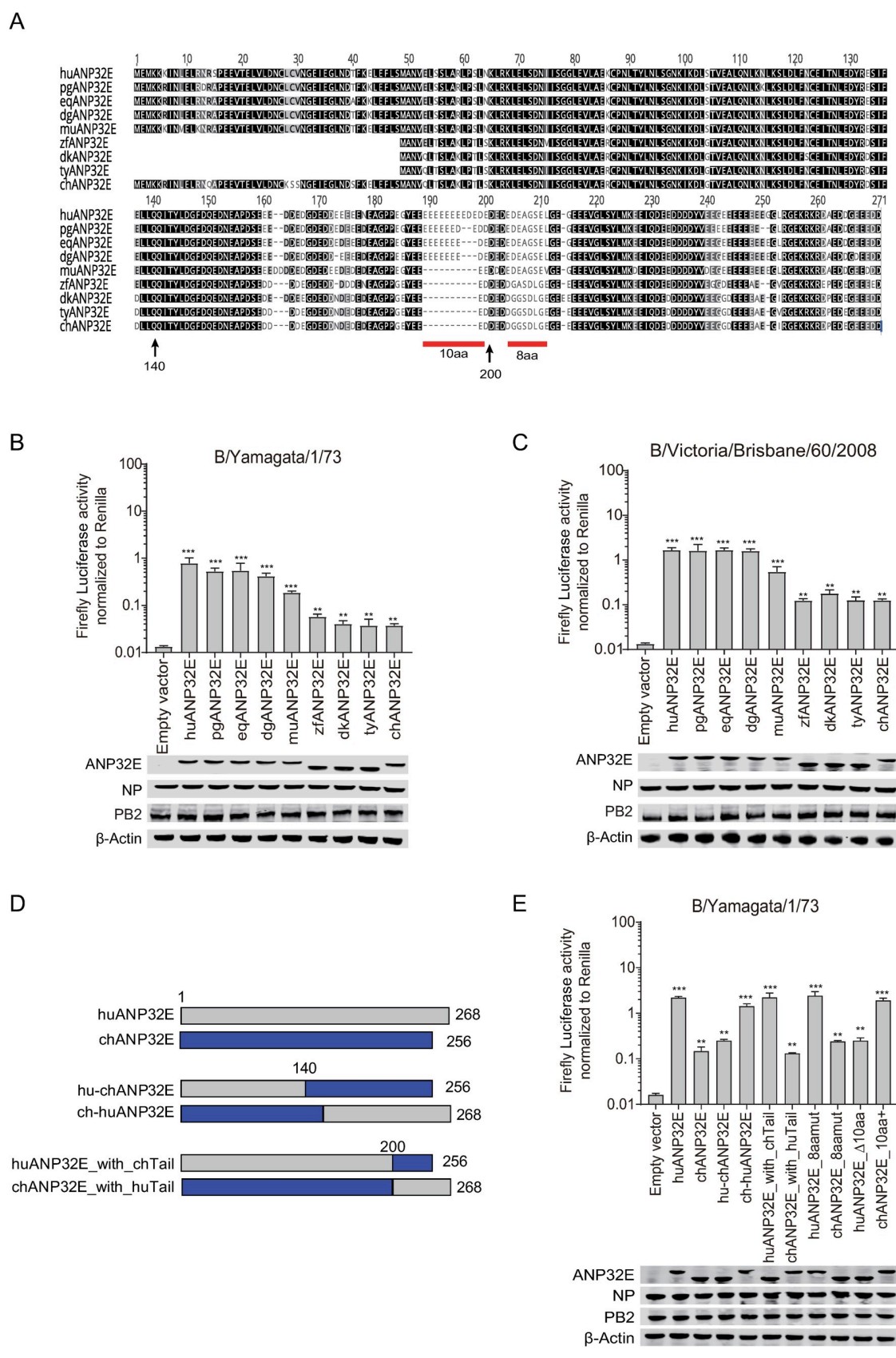

**Fig 5. Support of influenza B viral replication by ANP32E from different species and the key amino acids responsible for the support.** (**A**). The protein sequences of ANP32E for human (huANP32E), pig (pgANP32E), equine (eqANP32E), dog (dgANP32E), mouse (muANP32E), zebra finch (zbANP32E), duck (dkANP32E), turkey (tyANP32E) and chicken (chANP32E) were aligned using the Geneious R10 software. huANP32E was set as the reference sequence, and colors represent similarity of amino acid identity (Black = 100%, dark grey = 80–100%, light grey = 60–80%, white = <60%). Gaps are represented by dashes. Residue numbers correspond to huANP32E. TKO cells were co-transfected with 10 ng of ANP32E from different species or empty vector was co-transfected with minigenome reporter, *Renilla* expression control, and influenza B virus polymerase from B/Yamagata/1/73 (**B**), or B/Victoria/Brisbane/60/2008 (**C**). (**D**) Schematic diagram of the chimeric clones constructed between chicken and human ANP32E. The colors of the bars show the origins of the genes as follows: grey, huANP32E; blue, chANP32E. (**E**) Human or chicken ANP32E, or one of the chimeric clones were co-transfected with minigenome reporter, *Renilla* expression control, and B/Yamagata/ 1/73 polymerase into TKO cells. The expression of ANP32 proteins and polymerase was assessed using western blotting. Luciferase activity was measured 24 h later. The data indicate the firefly activity normalized to *Renilla*, Statistical differences between cells are labeled according to a one-way ANOVA followed by a Dunnett's test (NS = not significant, **P < 0.01, ***P < 0.001, ****P < 0.0001). Error bars represent the SD of the replicates within one representative experiment.

mammalian ANP32E. While muANP32E has a double aspartic acid insert at position 162–163, which is absent in other mammalian and avian ANP32E. We found that muANP32E has lower support to IBV polymerase then that of the other mammal (Fig 5A–5C), and the reason need to be further studied.

## A single amino acid at position 129 determines the support of ANP32Es to IBV polymerase

The above results showed that the activity of huANP32E in supporting IBV polymerase activity was 10 times lower than that of huANP32A or huANP32B (Fig 2A), and the avian ANP32Es have even lower support for IBV than huANP32A&B or those from other mammals (Fig 5B and 5C). Previous work demonstrated that chicken ANP32B provided no support for IAV polymerase because it harbors a 129I/130N signature which is a natural mutation away from the functional 129N/130D signature in other ANP32B molecules [14, 32]. By comparing the sequences, we found that huANP32E and chANP32E both have a Glutamic Acid (E) at amino acid position 129, which is conserved in ANP32E in most mammalian and avian species, while there is an Asparagine (N) in that position in ANP32A and ANP32B in mammals (Fig 6A). To investigate the impact of this 129E residue in ANP32E on its support of IBV polymerase, we generated two mutants, huANP32E_E129N and chANP32E_E129N. We found that in our co-transfection experiments, the support of huANP32E_E129N for IBV polymerase was significantly higher than that of huANP32E, and reached a level comparable to that of huANP32A; while the support of chANP32E_E129N for IBV polymerase activity was significantly higher than that of either chANP32E or chANP32A, and was not different from that of huANP32A (Fig 6B). These results suggest that the 129E is responsible for the low ability of chANP32E and huANP32E to support the viral polymerase.

## Different abilities of chicken and human ANP32A and ANP32E to bind to IBV polymerase

ANP32A and ANP32B are proposed to only bind to the complete IAV viral polymerase heterotrimeric complex, but do not bind the single subunit of polymerase to promote polymerase activity [12, 14, 15]. It has been suggested that the C-terminal acidic tail (LCAR) and amino acids 129–130 are important in the maintenance of the interaction between ANP32B and the influenza viral polymerase [14]. Whether either ANP32A or ANP32B can bind to the IBV polymerase is unknown. We performed a co-immunoprecipitation assay between IBV polymerase and ANP32 proteins, and found that huANP32B co-immunoprecipitated with IBV polymerase, but that the huANP32B C-terminal deletion mutants huANP32B_165T and

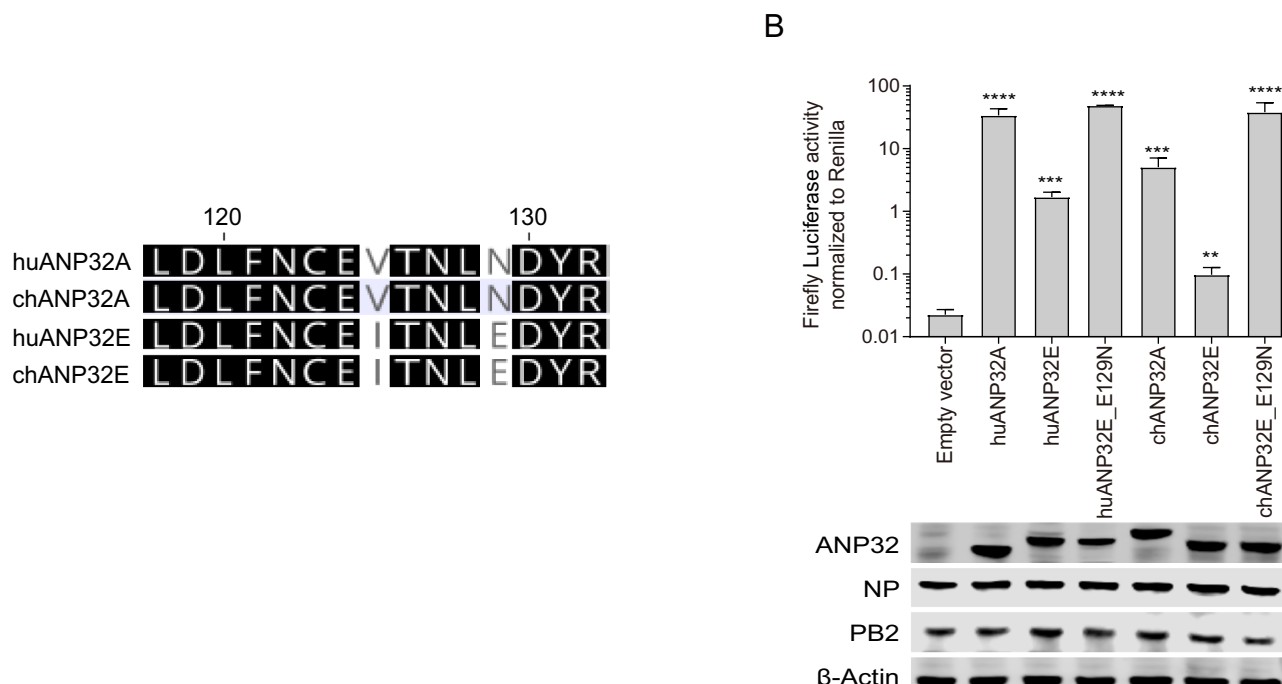

**Fig 6. A single amino acid at position 129 determines support of huANP32E and chANP32E for IBV polymerase.** (**A**) 129–130 site alignment of human ANP32A (huANP32A), chicken ANP32A (chANP32A), human ANP32E (huANP32E) and chicken ANP32E (chANP32E) protein sequences by Geneious R10 software. (**B**) Human and chicken ANP32 plasmids and different chimeric clones were co-transfected with minigenome reporter, *Renilla* expression control, and B/Yamagata/1/73 polymerase into TKO cells. The expression of ANP32 proteins and polymerase was assessed using western blotting. Luciferase activity was measured 24 h following transfection. The data indicate the firefly activity normalized to *Renilla*, Statistical differences between cells are labeled according to a one-way ANOVA followed by a Dunnett's test (NS = not significant, **P < 0.01, ***P < 0.001, ****P < 0.0001). Error bars represent the SD of the replicates within one representative experiment.

huANP32B_190T cannot bind with the viral polymerase; and the huANP32B_165T does not have the ability to support IBV polymerase activity, while huANP32B_190T has weak support (Fig 7A and 7B). However, huANP32B_216T, in which 35 amino acids were deleted, retained the ability to bind and support IBV viral polymerase (Fig 7A and 7B). When we compared the abilities of huANP32B, chANP32B, and a huANP32B with the functionally inactive mutations N129I/D130N, to bind and support the IBV polymerase, we found that chANP32B and huANP32B_N129I/D130N showed weaker binding ability to the IBV polymerase compared with that of huANP32B (Fig 7C), which is consistent with the polymerase activity assay. HuANP32A also showed strong binding to the IBV polymerase (Fig 7D). Inconsistent with the polymerase assay results, in which chANP32A gave lower support to the IBV polymerase than did huANP32A (Fig 3D and 3E), chANP32A and huANP32A$_{+33}$ showed similar binding to IBV polymerase to that of huANP32A (Fig 7D), suggesting that the immunoprecipitation results are not correlated with the viral polymerase activity.

We further confirmed that huANP32E, chANP32B, and chANP32E have lower binding ability to IBV polymerase compared that of huANP32A, huANP32B, and chANP32A. The E129N mutations in huANP32E and chANP32E enhance the binding to IBV polymerase, while with or without the 10-amino-acid insert in huANP32E or chANP32E could not affect their interaction with IBV polymerase (Fig 7E). These data are consistent with the supporting activity of different ANP32Es to IBV polymerase.

To investigate and characterize the interaction dynamics of huANP32A and chANP32A with the viral polymerase, we next carried out a surface plasmon resonance (SPR) assay to

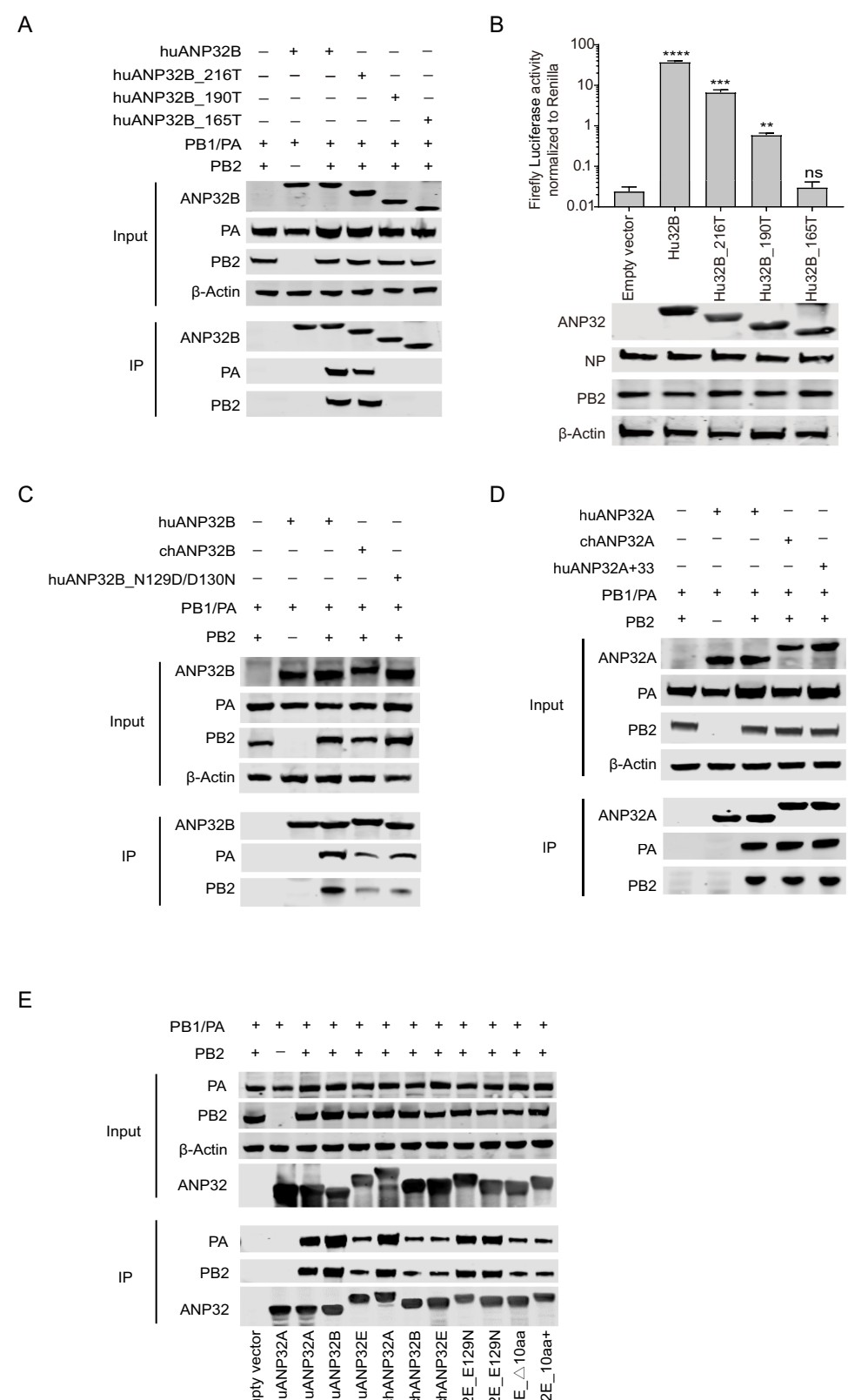

**Fig 7. Different binding abilities of chicken and human ANP32A and ANP32E for IBV polymerase.** (**A**) Detection of the interactions of differently truncated human ANP32B proteins with IBV polymerases. 293T cells were transfected with different truncated human ANP32B-Flag constructs, together with the viral polymerase subunits PA, PB1, and PB2. The coimmunoprecipitation of the anti-Flag antibodies and the proteins was assessed using western blotting. (**B**) Human ANP32B or its differently truncated clones were co-transfected with minigenome reporter, *Renilla* expression control, and B/Yamagata/1/73 polymerase into TKO cells. The expression of ANP32 proteins and polymerase was assessed using western blotting. Luciferase activity was measured 24 h following transfection. The data indicate the firefly activity normalized to *Renilla*, Statistical differences between cells are labeled according to a one-way ANOVA followed by a Dunnett's test (NS = not significant, **P < 0.01, ***P < 0.001, ****P < 0.0001). Error bars represent the SD of the replicates within one representative experiment. (**C to E**) 293T cells were transfected with different ANP32-Flag constructs, together with the viral polymerase subunits PA, PB1, and PB2. The coimmunoprecipitation of the anti-Flag antibodies and the proteins was assessed using western blotting. Detection of the interactions of human ANP32B (huANP32B), chicken ANP32B (chANP32B) and huANP32B with N129I/D130N mutations (huANP32B_N129I/D130N) proteins with IBV polymerases (**C**). Detection of the interactions of human ANP32A (huANP32A), chicken ANP32A (chANP32A), and human ANP32A with 33-amino-acid insert (huANP32A$_{+33}$) proteins with IBV polymerases (**D**). Detection of the interactions of human ANP32A (huANP32A), human ANP32B (huANP32B), human ANP32E (huANP32E), chicken ANP32A (chANP32A), chicken ANP32B (chANP32B), chicken ANP32E (chANP32E), human ANP32E with E129N mutation (huANP32E_E129N), chicken ANP32E with E129N mutation (chANP32E_E129N), human ANP32E with 10-amino-acid delete (huANP32E_△10aa) and chicken ANP32E with 10-amino-acid insert (chANP32E_10aa+) proteins with IBV polymerases (**E**).

evaluate the binding kinetics and affinity between polymerase and ANP32 proteins. HuANP32A, chANP32A, huANP32A+33, and huANP32A_165T were fusion expressed at downstream of GST-HRV3C peptide in a pCAGGS vector and purified using Glutathione Sepharose 4B and then digested by PreScission Protease. The purity of ANP32 proteins was checked using SDS-PAGE analysis and western blotting (S5 Fig). IBV polymerase PB1, PB2 and PA-His were expressed in 293T cells and the expression was checked by Ni Sepharose purification followed by SDS-PAGE analysis and mass spectrometry identification (S5 Fig).

Consistent with our co-IP results, all these three ANP32A proteins (huANP32A, chANP32A and huANP32A$_{+33}$) at concentrations of 0.15625–5 µM had valid binding stability to the viral polymerase immobilized on the CM5 chip in a dose-dependent manner (Fig 8A–8C). In contrast, huANP32A_165T has almost no binding to the polymerase (Fig 8D). In another control experiment, in which only two polymerase subunits (PA and PB1) were immobilized on the CM5 chip, no specific affinity was detected between huANP32A and the viral polymerase subunits (Fig 8E). However, the calculated dissociation constants (KD) for huANP32A, chANP32A and huANP32A$_{+33}$ from the SPR assay showed significant differences. In the SPR assay, the binding stability is highly related to the KD value, with the smaller value, the more stable the interaction. We found that the huANP32A binds to the IBV viral polymerase with a KD = 0.05418 µM, almost 3-fold lower than that of chANP32A. The KD value for huANP32A and polymerase binding also increased from 0.05418 µM to 0.1013 µM for huANP32A with the 33-amino-acid insert (Fig 8F). Although huANP32A, chANP32A and huANP32A$_{+33}$ have similar dissociation rate constants (kd), the association rate constants (ka) were distinctly different. The ka of huANP32A was twice as high as those from the other two proteins. This result indicated that huANP32A has a stronger binding affinity to IBV polymerase than chANP32A, and that the 33-amino-acid insert in chANP32A reduces the binding affinity of ANP32A to IBV polymerase. These results may help to explain why chANP32A gives only low support to the IBV polymerase.

## Discussion

Both IAV and IBV belong to the family of Orthomyxoviruses and are the two main types of influenza virus that cause epidemical infection in humans every year. IAV has been investigated extensively because it can cause both seasonal infection and pandemics. However, although accumulating evidence shows that IBV is also an important pathogen that causes

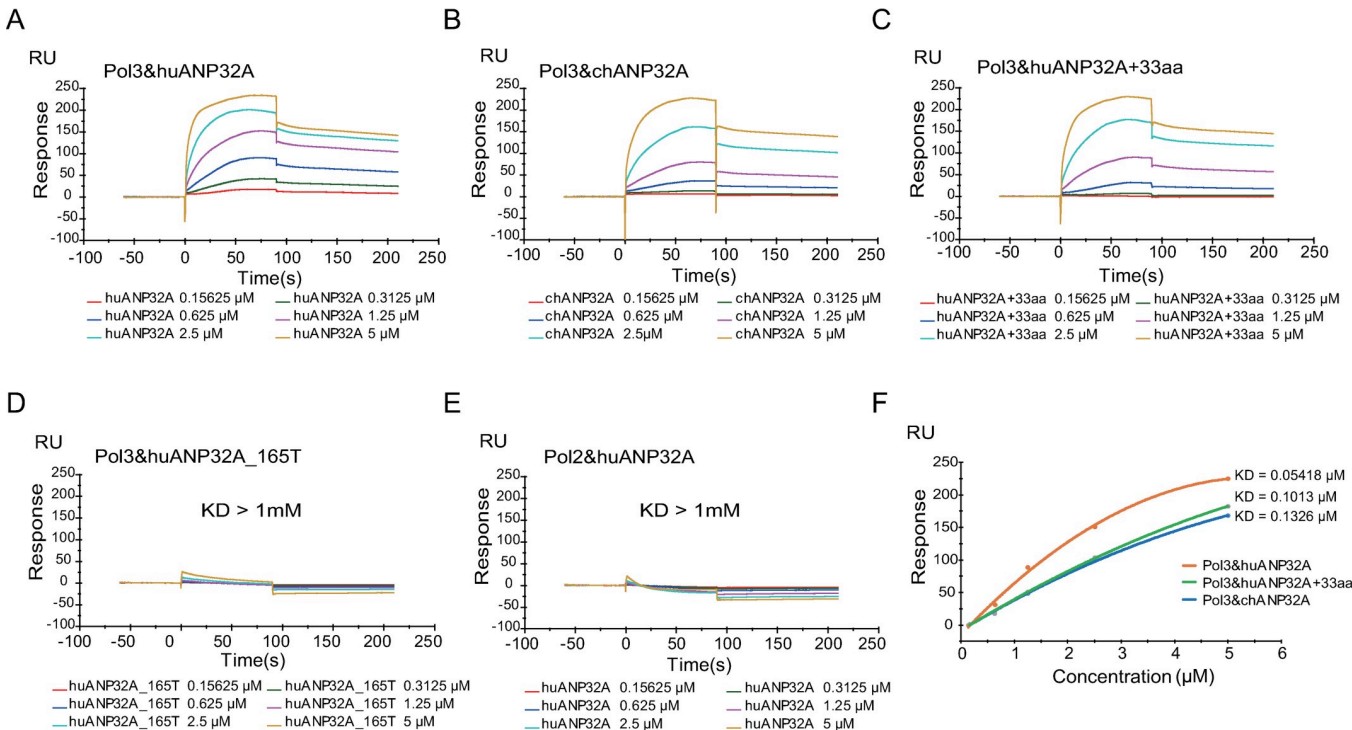

**Fig 8. The 33-amino-acid insert impacts the interaction dynamics between ANP32A proteins and viral polymerase.** (**A to E**) Surface plasmon resonance (SPR) measurements of the binding between IBV polymerase trimeric complex (Pol3) and the ANP32 proteins or their mutants purified from 293T cells. human ANP32A (huANP32A) (**A**). chicken ANP32A (chANP32A) (**B**). human ANP32A with the 33-amino-acid insert (huANP32A$_{+33}$) (**C**). human ANP32A C terminal truncated (huANP32A_165T) (**D**). human ANP32A (huANP32A) with 2 polymerase subunits (Pol2) as negative control (**E**). Different concentrations of ANP32 proteins were capture by the chips and shown are the corresponding sensor grams expressed in RU (response unit) versus time after subtracting the control signal. (**F**) Response units plotted against protein concentrations. Orange, huANP32A; green, huANP32A$_{+33}$; blue, chANP32A. The binding affinity (KD) values were calculated using a 1:1 fit model produced with Biacore T200 analysis software (Biacore T200 Evaluation Software Version 3.1).

high morbidity and mortality in human populations, the characters and replication mechanism of IBV remain largely unknown. In this study, we demonstrate that the polymerase activity of IBV depends on the human cellular ANP32 proteins, of which ANP32A and ANP32B gave strong support to IBV polymerase activity. ANP32E, a member of ANP32 family that was shown no function in supporting IAV polymerase, gave mild support to IBV polymerase activity. Avian ANP32 proteins gave weak support to IBV replication, which we demonstrate to be because they harbor certain mutations compared with these proteins in mammals. These results revealed the importance of ANP32 proteins in IBV replication and the distinct species-specific usage of ANP32 proteins of IAV and IBV polymerase.

The ANP32 family comprises several members, including ANP32A, ANP32B, ANP32C, ANP32D, and ANP32E. ANP32A, ANP32B, and ANP32E are conserved in vertebrates, but the ANP32C and ANP32D are predicted to be intronless-gene-coded proteins, therefore, their genes are considered to be pseudogenes or retrogenes [29]. In a previous study, we found that ANP32A and ANP32B provide fundamental support to influenza A virus replication, and that double knockout of ANP23A and ANP32B (DKO) aborted polymerase activity of IAV [14]. Here, we show that a single knockout of ANP32A, ANP32B, or ANP32E does not affect IBV polymerase activity. Interestingly, double knockout of ANP32A and ANP32B reduces IBV polymerase activity by 100-fold, which is not as strong an effect as that on the IAV polymerase (which reduced more than 1000 folds in DKO cells), suggesting that another factor may

contribute to IBV polymerase activity. As expected, triple knockout of ANP32A, ANP32B, and ANP32E (TKO) further reduces IBV polymerase activity by 10-fold (Fig 1). Reconstitution of ANP32A or ANP32B can fully restore IBV polymerase activity, but reconstitution of ANP32E can only partially restore IBV polymerase activity, suggesting that ANP32E has only limited ability to support IBV replication (Fig 2). We confirm that ANP32E does not support IAV polymerase activity.

All our evidence supports the argument that, similar to IAV, IBV may rely on ANP32A or ANP32B in its replication, with the exception that IBV can also use ANP32E to a certain extent. It is very interesting to see that the amino acid at position 129 of ANP232E is Gluta-mine acid (129E), while that in the corresponding position of ANP32A or ANP32B is Aspara-gine (129N). An E129N mutation in ANP32E can completely restore support of IBV polymerase activity (Fig 6), indicating that the 129E of ANP32E is a dominant impact on its ability to support IBV. We also observed that the ability of avian ANP32E in supporting IBV polymerase is more than 10 times lower than that of huANP32E, which was due to a 10-amino-acid deletion in the LCAR domain. This species-specific difference was due to a 10-amino-acid deletion in the LCAR domain of avian ANP32E.

Unlike IAV, which comprises a heterogeneous group of different subtypes, IBV forms a more homogeneous cluster with two main lineages, Victoria and Yamagata. IAV can infect many animal species but IBV can only be identified clinically in humans, seals, and pigs. Infection by IBV in avian species has not been confirmed. Taking into consideration that IBVs have the ability to bind to α2,6-linked or α2,3-linked sialyl-glycans [33], it is proposed that there are interspecies barriers existing, protecting the avian hosts against IBV infection. To further illustrate the contribution of ANP32 proteins to IBV replication and possible interspecies restriction, we compared the abilities to support IBV replication by ANP32A, ANP32B, and ANP32E proteins from different species. Chicken ANP32A, ANP32B, or ANP32E showed related low activity to support IBV polymerase in TKO or chicken DF-1 cells compared with human ANP32A and ANP32B. It is intriguing to find out that chicken ANP32A gives only weak support to IBV polymerase because of the extra 33- or 29-amino-acid inserts (Fig 3). The 33-amino-acid insert is required for chANP32A to support chicken IAV polymerase activity, while proteins with this insert are unable to support IBV replication. This result indicates different evolutionary patterns of IAV and IBV in the use of host ANP32 proteins. It is worth to note that avian ANP32A has three isoforms due to differential splicing, including a long iso-form with 33-amino-acid insert in exon 4, a shorter isoform with 29-amino-acid insert, and a mammalian-like isoform without insertion [12, 15]. The express ratios of the different iso-forms in avian cells are varied. The abundance of the mammalian-like isoform of chANP32A in DF-1 cells is about 9% in all expressed isoforms, and the 33 insert and 29 insert isoforms are 66% and 25% respectively [15]. We identified that overexpressing of huANP32A or huANP32B in DF1 cells enhances IBV polymerase activity and viral replication, indicating a limited support to IBV polymerase exists in DF-1 cells. We also found that when we use a high MOI of B/Yamagata/PJ/2018 virus to inflect DF1 cells, the virus can replicate to a certain level (Fig 2C). It remains unknown whether the avian species with high ratio of mammalian-like isoform of chANP32A expression can support better IBV replication.

It is known that the IAV polymerase complex binds to ANP32A or ANP32B for its normal function. The chANP32A shows stronger binding ability to avian IAV strains than do mammal ANP32As or ANP32Bs, which is consistent with its ability to support chicken IAV replication. The 33-amino-acid insert is responsible for this stronger binding ability [13]. Surprisingly, we found that chANP32A and huANP32A showed a similar ability to bind IBV polymerase, despite the fact that they showed dramatically different abilities to support the activity of IBV polymerase. It is not known to date how the ANP32 proteins interact with the

polymerase, or how this interaction between ANP32A proteins and viral polymerase can support viral polymerase replication. Our experiments show that although the huANP32A and chANP32A have similar binding ability to IBV polymerase in co-IP assay, the SPR assay shows that the association rate constant (ka) of huANP32A is higher than that of chANP32A, therefore, the calculated dissociation constant (KD) of huANP32A to the polymerase is significantly lower than that of chANP32A, indicating that huANP32A has a higher affinity to the polymerase and thus may lead to a stronger ability to support the activity of the polymerase of IBV. Given the fact that the viral polymerase have highly compact structure and interact with many host factors during replication [17, 34], it is remain unclear that the detailed interaction mechanism between ANP32A and polymerase of IBV and how much this interaction contributes to the host range selection.

Taking the roles of ANP32 proteins in IAV and IBV together, we summarized the current understanding of interaction between ANP32 proteins and influenza viral polymerase. HuANP32A and huANP32B play major role in supporting both human IAV and IBV replication, but huANP32E has mild support to IBV and no support to human IAV. None of the human ANP32A, ANP32B, or ANP32E supports avian IAV because they do not have a 33-amino-acid insert as chANP32A. Chicken ANP32A harbors a 33-amino-acid insert which enables supporting to both avian and mammal IAV, but not IBV. Chicken ANP32B is a nature inactive molecule to IAV and IBV polymerase because of 129I/130N substitution. Chicken ANP32E has no function in supporting IAV polymerase, but shows a weak support to IBV. The low level support to IBV of ANP32E is due to a 129E mutation. Thus, none of chicken ANP32 proteins show good support to IBV (Fig 9). The differences in using of the ANP32 proteins by IAV and IBV are correlated to the species-specific restriction of influenza virus replication. In conclusion, our functional investigation of ANP32 proteins in supporting IBV replication and the species-specific restriction of IBV polymerase may provide new insight to understand viral adaptation and evolution.

## Materials and methods

Human 293T (ATCC CRL-3216) and MDCK (CCL-34) cells were cultured with Dulbecco's modified Eagle's medium (DMEM, Sigma) supplemented with 10% fetal bovine serum (FBS; Clark) and 1% penicillin and streptomycin (Gibco). The polymerase plasmids of human influenza A virus H7N9 A/Anhui/01/2013 (H7N9$_{AH13}$) were kind gifts from Dr. Hualan Chen. The plasmids carrying the genes of influenza Bvirus B/Yamagata/1/73 were kindly provided by Dr Yoshihiro Kawoaka. B/Yamagata/PJ/2018 (available in our lab, GenBank accession numbers: NP: MN700018, PB1: MN700019, PB2: MN700020, PA: MN700021) was used to infect 293T and knockout cells. pCAGGS plasmids containing ANP32A, ANP32B from different species are kept in our lab [18]. The pCAGGS plasmids containing full length ANP32E isoforms of several species and influenza B virus B/Victoria/Brisbane/60/2008 were generated by gene synthesis (Synbio technologies, China) according to the sequences deposited in GenBank, including human ANP32E (huANP32E, NM_030920.5, NP_112182.1), pig ANP32E (pgANP32E, XM_021089919, XP_020945578.1), equine ANP32E (eqANP32E, XM_001917235.4, XP_001917270.1), dog ANP32E (dgANP32E, XM_003639621.4, XP_003639669.1), mouse ANP32E (muANP32E, NM_023210.4, NP_075699.3), zebra finch ANP32E (zfANP32E, XM_012570886.1, XP_012426340.1), duck ANP32E (dkANP32E, XM_005030153.3, XP_005030210.2), turkey ANP32E (ty ANP32E, XM_003212772.3, XP_003212820.2), chicken ANP32E (chANP32E, NM_001006564.2, NP_001006564.2), huANP32E_with_chTail, chANP32E_with_huTail, huANP32E_8aamut, chANP32E_8aamut, huANP32E$_{\Delta10}$, chANP32E$_{+10}$, B/Victoria/Brisbane/60/2008 PB1 (CY115157.1, AFH57918.1), B/Victoria/Brisbane/60/2008

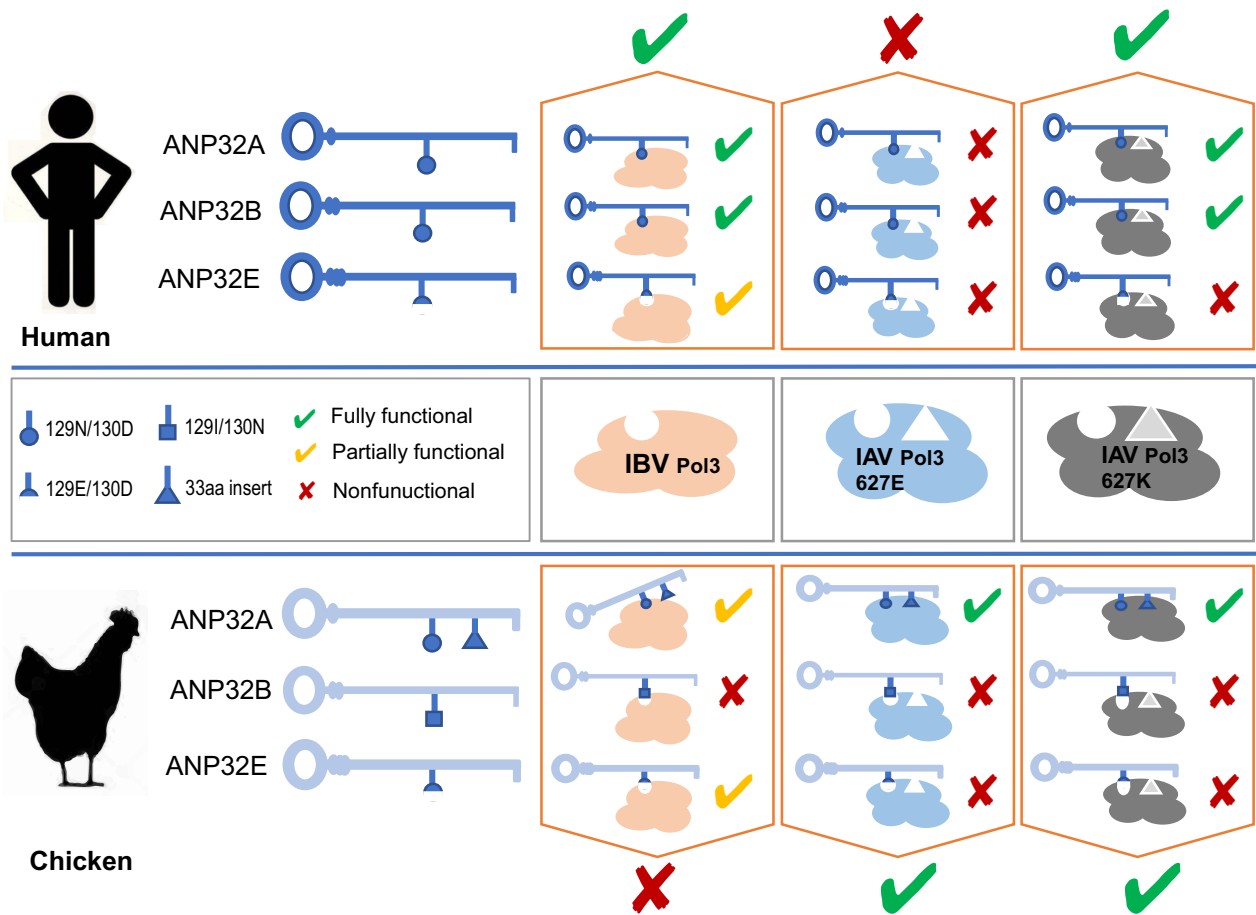

**Fig 9. Selective usage of ANP32 proteins by IAV and IBV polymerase and their molecular basis.** Schematic model of interaction between ANP32 proteins and influenza viral polymerase. ANP32A, ANP32B, ANP32E from human and chicken interact with polymerase (Pol3) of IBV, avian-origin IAV (with PB2 627E), and mammal adapted IAV (with PB2 627K). The specific amino acid residues are indicated.

PB2 (CY115158.1, AFH57919.1), B/Victoria/Brisbane/60/2008 PA (CY115156.1, AFH57917.1), B/Victoria/Brisbane/60/2008 NP (CY115154.1, AFH57914.1). To obtain pCAGGS- Yamagata-PB2-Flag and pCAGGS-Victoria-PB2-Flag plasmids, pHH21-Yamagata-PB2 and pCAGGS-Victoria-PB2 was used as the template to amplify the PB2-Flag sequences, and then fused with pCAGGS vector according to the online In-Fusion HD Cloning Kit User Manual (http://www.clontech.com/CN/Products/Cloning_and_Competent_Cells/Cloning_Kits/xxclt_searchResults.jsp). Using the same method, we changed the promoter of the firefly minigenome reporter from human to chicken to generate pchPOL1-vluc. To create the pCAGGS-huANP32A$_{+33}$ plasmid, pCAGGS-chANP32A was used as the template to amplify the 33 amino acids, and then fused with the pCAGGS-huANP32A. Site-directed mutants of these sequences were generated using overlapping PCR and identified using DNA sequencing.

## Knockout cell lines

The generation of the 293T AKO, BKO and DKO knockout cell lines were described in our previous report [14]. EKO and TKO knockout cell lines were generated using the same approach. Briefly, 293T or DKO cells cultured in 6-well plates were transfected with 0.5 μg pMJ920 (Addgene plasmid # 42234) plasmids and 0.5 μg gRNA expression plasmids in Polyet

Transfection Reagent (Signagen, SL100688) using the recommended protocols. GFP-positive cells were sorting by flow cytometry (MoFlo XDP, Backman) at 24 h post-transfection, then monoclonal knockout cell lines were screened using western blotting and/or DNA sequencing.

## Polymerase assay

HEK293T or DF1 cells were transfected with plasmids for the expression of the viral proteins PB1, PB2, PA, NP and pPol1-WSN-HAutr-vluc/pPol1-B/Yamagata/1/73-NSutr-vluc or pchPol1-B/Yamagata/1/73-NSutr-vluc. *Renilla* luciferase expression plasmids (pRL-TK, kindly provide by Dr. Luban*)* were used as an internal control for the dual-luciferase assay. To determine the effect of ANP32 proteins on viral polymerase activity, 293T or DF1 cells in 24-well plates were transfected with plasmids of PB1 (20 ng), PB2 (20ng), PA (10 ng) and NP (40 ng), together with 40 ng vluc and 5 ng *Renilla* luciferase expression plasmids, using Polyjet Transfection Reagent (Signagen, SL100688) according to the manufacturers' instructions. As a negative control, cells were transfected with the same plasmids, with the exception of the PB2 or ANP32 expression plasmid. After transfection, the cells were incubated at 37˚C for 24 h, and then luciferase activity was measured with a dual-luciferase reporter system (Promega, E1960) on a Centro XS LB 960 luminometer (Berthold technologies) according to the manufacturer's instructions. The expression levels of polymerase proteins in different cell lines were detected by western blotting, using specific antibodies (Genetex, GTX128538) for NP and anti-Flag tag antibody (Sigma, F1804) for PB2-Flag protein.

## Influenza virus infection and infectivity

HEK293T cells, huANP32A knockout 293T cells (AKO cells), huANP32B knockout 293T cells (BKO cells), huANP32A &B double knocked out 293T cells (DKO), huANP32A, huANP32B, and huANP32E triple knocked out 293T cells (TKO), huANP32B or empty vector transfected DF1 cells were infected with B/Yamagata/PJ/2018 virus at a multiplicity of infection (MOI) of 0.1 for 2 h, washed twice with PBS, and then cultured at 37˚C in Opti-MEM containing tosyl-sulfonyl phenylalanyl chloromethyl ketone (TPCK)-trypsin (Sigma) at 0.5 μg/ml. At the indicated time points, the culture supernatant was harvested and a Focus Formation Units Assay (FFU) was run as previously described [35].

## Immunoprecipitation and western blotting

For immunoprecipitation and western blotting, transfected cells were lysed using an ice-cold lysis buffer (50 mM Hepes-NaOH [pH 7.9], 100 mM NaCl, 50 mM KCl, 0.25% NP-40, and 1 mM DTT), and centrifuged at 13,000× g and 4˚C for 10 min. After centrifugation, the crude lysates were incubated with Anti-FLAG M2 Magnetic Beads (SIGMA-ALDRICH, M8823) at 4˚C for 2 h. After incubation, the resins were collected by magnetic separator and washed three times with PBS. The resin-bound materials were eluted by 3X Flag peptide (150 ng/ul) and subjected to SDS-PAGE, then transferred onto nitrocellulose membranes. Membranes were blocked with 5% milk powder in Tris-buffered saline (TBS) for 2 h. Incubation with the first antibody (Anti-Flag antibody from SIGMA (F1804), Anti-NP antibody from Genetex (GTX128538), Anti-β-actin from Sigma(A1978)) was performed for 2 h at room temperature (RT), followed by washing three times with TBST. The secondary antibody (KPL, 1:10,000) was then applied and samples were incubated at RT for 1 h. Subsequently, membranes were washed three times for 10 min with TBST. Signals were detected using a LI-COR Odyssey Imaging System (LI-COR, Lincoln, NE, USA).

## Surface plasmon resonance (SPR) measurement

The binding activity of different ANP32 proteins to the IBV polymerase was measured using a Biacore T200 instrument (GE Healthcare). Anti-His antibody was immobilized on the CM5 chip surface (flow cells 1 and 2) via the amine coupling method. 293T cells transfected with PA-His, PB1, with or without PB2 were lysed with cell lysis buffer (50 mM Hepes-NaOH [pH 7.9], 100 mM NaCl, 50 mM KCl, 0.25% NP-40, and 1 mM DTT). Cells lysates were diluted with running buffer (HBS-EP+) to the corresponding concentration and allowed to flow through the immobilized chip for 90 s 5 ul/min. ANP32 proteins were individually fusion expressed at downstream of GST-HRV3C peptide in a pCAGGS vector backbone in DKO cells. Proteins were purified using Glutathione Sepharose 4B and then digested by PreScission Protease (Beyotime, P2303). The purified proteins were diluted with running buffer to different indicated concentrations and control flow cells at a flow rate of 30 μl/min for 90s. After 120s dissociation, the chip surface was regenerated with 10mM Glycine-HCL pH 1.5 at 30 μl/min for 45 s. Data were analyzed using Biacore Evaluation 3.1 software with a 1:1 fit model.

## Statistics

Statistical analyses were performed using GraphPad Prism, version 7.04 (Graph Pad Software, USA). Statistical differences between groups were assessed using One-way ANOVA followed by a Dunnett's post-test. Error bars represent the SD (standard deviation) of the replicates within one representative experiment. NS, not significant ($p > 0.05$), *$p < 0.05$, **$p < 0.01$, ***$p < 0.001$, ****$p < 0.0001$. All the experiments were performed independently at least three times.

## Supporting information

**S1 Fig. Viability of TKO cells measured by CCK-8 assay.** The cell viability of 293T and TKO cells were measured at 24, 48, and 72 h by the CCK-8 reagent in accordance with the manufacturer's instructions (Beyotime Biotechnology, Shanghai, China).
(PDF)

**S2 Fig. Replication of influenza B virus in MDCK and 293T cells.** MDCK and 293T cells were infected with B/Yamagata/PJ/2018 virus at a MOI of 0.1. The supernatants were sampled at 12, 24, 36, 48, 60, and 72 h post infection and the viral titers were determined using Fluorescence Focus Units (FFU) assay on MDCK cells. The result is shown as average of $n = 3 \pm SD$.
(PDF)

**S3 Fig. ANP32 proteins supported the IBV viral polymerase activity in a dose-dependent manner.** Increasing doses of huANP32A(A), huANP32B(B) or huANP32E(C) were co-transfected with minigenome reporter, Renilla expression control, influenza B virus polymerase of B/Yamagata/1/73 in TKO cells. The expression of ANP32 proteins and polymerase was assessed by western blotting. Luciferase activity was measured 24 h later. (Data are firefly activity normalized to Renilla, Statistical difference between cells were labeled, according to a one-way ANOVA followed by a Dunnett's test; NS = not significant, *$P < 0.05$, **$P < 0.01$, ***$P < 0.001$, ****$P < 0.0001$. The results represent at least three independent experiments.)
(PDF)

**S4 Fig. Sequence alignment of ANP32A and ANP32B proteins from different species.** The protein sequences of ANP32A for human (huANP32A), pig (pgANP32A), equine (eqANP32A), dog (dgANP32A), ostrich(osANP32A), zebra finch (zbANP32A), duck (dkANP32A), turkey (tyANP32A), and chicken (chANP32A) were aligned using the Geneious

R10 software. huANP32A was set as the reference sequence. The colors represent similarity of amino acid identity (Black = 100%, dark grey = 80–100%, light grey = 60–80%, white = <60%). Gaps are represented by dashes. Residue numbers correspond to huANP32A.
(PDF)

**S5 Fig. Purification and identification of ANP32 proteins and viral polymerase. (A)** ANP32 proteins were fusion expressed at downstream of GST-HRV3C peptide in a pCAGGS vector and purified using Glutathione Sepharose 4B and then digested by PreScission Protease. Purified ANP32 proteins were diluted to 100ug/ml and 1ug of the purified protein was checked using SDS-PAGE analysis and western blotting. **(B)** IBV polymerase PB1, PB2 and PA-His were expressed in 293T cells and purified with Ni Sepharose (GE). The purified protein was checked using SDS-PAGE analysis. **(C)** The proteins of the purified band in (B) were identified using the mass spectrometry.
(PDF)

## Acknowledgments

We thank Dr. Hualan Chen and Dr. Yoshihiro Kawaoka for providing plasmids. We thank Dr. Ervin Fodor for helpful discussions. We thank the Core Facility of the Harbin Veterinary Research Institute, the Chinese Academy of Agricultural Sciences for providing the technic support.

## Author Contributions

**Conceptualization:** Zhenyu Zhang, Xiaojun Wang.

**Data curation:** Zhenyu Zhang, Haili Zhang.

**Formal analysis:** Zhenyu Zhang, Xiaojun Wang.

**Funding acquisition:** Zhenyu Zhang, Xiaojun Wang.

**Investigation:** Zhenyu Zhang, Haili Zhang, Ling Xu, Xing Guo, Wenfei Wang, Yujie Ji, Chaohui Lin, Yujie Wang.

**Methodology:** Zhenyu Zhang, Haili Zhang, Wenfei Wang.

**Project administration:** Zhenyu Zhang, Xiaojun Wang.

**Supervision:** Xiaojun Wang.

**Validation:** Zhenyu Zhang, Haili Zhang.

**Writing – original draft:** Zhenyu Zhang, Xiaojun Wang.

**Writing – review & editing:** Zhenyu Zhang, Xiaojun Wang.

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
