## [Decision Letter · Decision Letter 0]

10 May 2020

Dear Dr Wang,

Thank you very much for submitting your manuscript "Selective usage of ANP32 proteins by Influenza B Virus Polymerase: implications in determination of host range" (PPATHOGENS-D-20-00687) for consideration at PLOS Pathogens. As with all papers peer reviewed by the journal, your manuscript was reviewed by members of the editorial board and by several independent peer reviewers. Based on the reports, we regret to inform you that we will not be pursuing this manuscript for publication at PLOS Pathogens

The reviews are attached below this email, and we hope you will find them helpful if you decide to revise the manuscript for submission elsewhere. We are sorry that we cannot be more positive on this occasion. We very much appreciate your wish to present your work in one of PLOS's Open Access publications. 

Thank you for your support, and we hope that you will consider PLOS Pathogens for other submissions in the future.

Sincerely,

Ron Fouchier

Section Editor

PLOS Pathogens

Kasturi Haldar

Editor-in-Chief

PLOS Pathogens

orcid.org/0000-0001-5065-158X

Michael Malim

Editor-in-Chief

PLOS Pathogens

orcid.org/0000-0002-7699-2064

Reviewer's Responses to Questions

**Part I - Summary**

Reviewer #1: The authors dissect species-specific functional interactions between ANP32 proteins and influenza viruses, with a focus on influenza B virus. They find that avian ANP32 molecules are generally unable to efficiently support influenza B virus polymerase activity, which is the opposite to avian influenza A virus. Using mutagenesis, binding assays and KO technologies, they map the specific determinants in different ANP32 molecules that are responsible for this effect. The work is notable for its use of constructs from many different species, and its fine dissection of different ANP32 genes (A/B/E) and isoforms. The finding that ANP32E plays a role in influenza B virus replication, but not ANP32A is a highlight. The data are of very high quality and clearly support the conclusion that avian cells may not be permissive for influenza B virus replication due to incompatibility with ANP32s. Taken together with the work on influenza A viruses, this new work is therefore very important as it suggests interesting species-specific evolution of influenza viruses with host ANP32 molecules.

Reviewer #2: Influenza A viruses (IAV) circulate in birds and infect many mammalian species. Although influenza A and B viruses are closely related, IBVs do not infect birds, and the cause of this host range restriction is unknown. ANP32A was shown to be the major species barrier preventing avian IAV polymerases from functioning in mammalian cells. Species-specific differences in ANP32A, ANP32B and ANP32E have since been shown to impact the function of the viral polymerase in different hosts. Here, Zhang, Zhang, et al. thoroughly evaluate the role of ANP32s in supporting influenza B virus polymerase activity. Using a clean genetic background (ANP32A, ANP32B, ANP32E triple knockout; TKO), the authors show that human ANP32A and B support IBV polymerase and that either protein is sufficient for viral replication. ANP32E does not support IAV polymerase, but the authors make the interesting observation that ANP32E provides low levels of activity for the IBV polymerase. None of the avian ANP32 proteins support IBV. Mutant and chimeras reveal that an insertion of acidic residues in human ANP32E contributes to its ability to support IBV polymerase. However, this activity is restricted by the glutamate encoded at amino acid 129; converting this to the asparagine found in ANP32A confers a high degree of activity to both human and avian ANP32E. Thus, ANP32E functions poorly due to variation at aa129, and the species-specific absence of an acidic stretch further eliminates activity in birds, suggesting ANP32E is a major factor restricting IBV host range. Physical interactions between IBV and ANP32A and B are demonstrated by coIP and SPR. The functional assays and dissection of species-specific activities are well-controlled and compelling. However, the authors over-interpret their SPR data, especially in light of how ANP32A interactions do not correlate with support of polymerase activity. Nonetheless, this submission offers strong evidence to explain why influenza B viruses do not circulate in infect birds.

Reviewer #3: Influenza B virus is the virus type that together with influenza A viruses (IAV) causes seasonal outbreaks of influenza. A distinctive feature of influenza B virus is the restriction to humans as the main host species, whereas only IAV has established phylogenetic stable lineages in other host species including birds and pigs. The molecular basis for the species specificity of influenza viruses is currently in the focus of intense research and pre-pandemic risc assessment. Previous work has identified the functional interaction of host proteins of the ANP32 family with the viral polymerase complex as a major determinant of host range. While human ANP32-A and -B proteins are essential co-factors of human-adapted polymerases, e.g. chicken ANP32A supports only avian polymerase activity. Several recent papers have unravelled these interactions, the contribution of different ANP32 proteins and the detailed molecular requirements on both viral and host proteins. In this manuscript, Zhang et al describe the requirement of influenza B virus for human ANP32A or B proteins to support polymerase activity and replication in human cells, while chicken orthologues fail to fully support these human-specific viruses. The authors then expand these analyses on a set of ANP32 proteins from other species and demonstrate that the molecular basis for this species specific interaction resides in a 33 amino acid stretch that is exclusive to chicken ANP32A. Further, they investigate the role of ANP32E, which is shown to only play a minor role in supporting IBV polymerase activity. Finally, binding of polymerase and ANP32 proteins is investigated.

The work is well-structured, follows a clear line of thought and is presented in an understandable way, although language would benefit from some enhancements. The conclusions are valid and the data are of some interest to the community. However, much of the work is rather confirmatory with incremental advance in our understanding of the role of ANP proteins in influenza B virus infection at this stage of analysis. This is in part also due to the strong focus of the work on the use of ectopically expressed genes and less on the situation in virus-infected cells.

1.) A major result of this manuscript, the dependence of influenza B virus propagation on ANP32A and -B proteins, has already been demonstrated (e.g. Staller et al., J.Virol. 2019). Due to its depth of analysis, the current manuscript extends this finding, but is rather confirmatory in character. The second novel aspect is the analysis of the role of ANP32E. As the authors show and state themselves, however, the contribution of this isoform to polymerase function is smaller compared to the previously identified major host range determinants ANP32A and B.

2.) The major body of evidence gained in this manuscript relies on standard polymerase activity assays. These are performed with skill and the immunoblots showing control expressions are of high quality. However on top of that, only one panel (Fig.1D) investigates the impact of ANP32 proteins on virus replication. Also, little significant novel insight is gained here, since deletion of ANP32E in cells already lacking ANP32A and B does not have an impact on virus replication. The interaction analyses in figures 7 and 8 again only marginally increase knowledge.

**Part II – Major Issues: Key Experiments Required for Acceptance**

Reviewer #1: No major issues identified

Reviewer #2: 1. Experiments studying interactions between IBV polymerase and ANP32s are underdeveloped and the claims need to be softened

a. The authors use results from SPR to calculate Kd and claim avian ANP32A binds IBV polymerase ~2.5 fold less well than human ANP32A. However, these data have no statistics associated with Kd calculations. The curve in Fig 8F does not correlate with the individual data in 8A-C. If anything, data in 8C suggest higher response rates for huANP32A+33 than for ANP32A. 

b. Minor differences in Kd determined by SPR can be caused by differences in protein quality or concentration. Data showing the purity and equivalent concentration of ANP32s is needed. It would also be useful to understand how well IBV polymerase proteins are expressed, as the methods say these proteins are captured on the chip directly from cell lysate.

c. While a major new claim is that ANP32E support IBV polymerases, ANP32E was not included in any of the binding assays. 

d. IBV polymerase interacts by coIP with both huANP32A and chANP32A, yet only huANP32A supports function. If the authors are able to confirm minor differences in binding proposed in Fig 8, would these even be relevant to IBV pol function in cells? 

e. Given the limited support of Fig 8, lines 388-399 in the Discussion are too speculative. In addition, SPR was done in the absence of viral RNAs, thus different functional states of the polymerase cannot be assumed.

2. The authors identify a 10aa insertion of acidic residues present in mammalian ANP32E that they nicely demonstrate is important for the low levels of huANP32E activity. While murine ANP32E is tested in cells, it is conspicuously absent from their alignment. 

a. Alignments show that muANP32E lacks this insert. This should be addressed in the text.

b. Fig 4A,B; it is not clear why muANP32B is used as a control here. Staller et al. 2019 showed that muANP32B actually encode 129S 130D, which is different than the more common 129N 130D found in huANP32B, huANP32A, and chANP32A. Fig 6 here goes on to show that the identify of aa129 is important for ANP32E function. This difference in muANP32E should also be addressed in the text. 

c. care should also be taken to make clear that the 10aa insertion confers species-specific function, while residue 129 is the dominant change controlling activity regardless of the host.

3. The authors claim that ANP32s are a major barrier preventing IBV replication in birds. However, this was never tested. Does expression of huANP32A or B (or perhaps chANP32E_E129N) allow IBV replication in birds, or do other barriers still exist?

Reviewer #3: (No Response)

**Part III – Minor Issues: Editorial and Data Presentation Modifications**

Reviewer #1: Lines 163-167. Please check this statement. I thought that mammalian ANP32s were ‘short’, while avian ones were ‘long’? Statement seems contrary to line 160.

Line 185. Please clarify what is meant by chickens have an ANP32A isoform that lacks the 4 hydrophobic amino-acids – are the authors referring to multiple isoforms of ANP32A? Please make this clearer – perhaps discuss different isoforms earlier in the introduction to make it obvious to the reader that they exist. Also, chickens are reported to have a short isoform lacking the entire ‘insertion’ – eg like del33. Please comment.

Line 188 – please specific what ‘alter’ means – enhance, reduce? Not clear what meant here.

Line 193 – other than chickens, some birds have higher ratios of the ‘short’ ANP32A isoform to the ‘long’ isoform (Baker et al, Domingues et al) – can the authors make a definitive statement whether these birds could support IBV replication? In the discussion, perhaps comment on the amino-acid compositions in ANP32B and ANP32E of the bird species that are dominant in expressing ANP32Adel33 isoforms. Would their ANP32B/E molecules also suggest a support for IBV polymerase?

Figure 4 – what is muANP32B? Murine? Why is its apparent size larger than huANP32B? Doesn't seem to be referred to anywhere in the text of Figure legend.

Reviewer #2: 1. Fig legends 3, 4, 5B,C, do not indicate which cell lines are being used. This reviewer assumes the experiments are in TKO. If not, I will have to reconsider my interpretation of the data.

2. A model or table in the Discussion indicating the functionality of avian and human ANP32s with huIAV, chIAV or huIBV would be a useful addition to help solidify current and previous findings.

3. Table 1 was not present in the manuscript received for revue. However, given the concerns about SPR above, it is not clear how much Table 1 contributes.

4. Please indicate that Staller et al. 2019 showed that ANP32B enhances influenza B polymerase activity.

5. Statistics throughout. Reporting SEM for technical replicates is not accurate, please use SD to illustrate error bars and to run statistical tests on. Number of replicates are not indicated.

6. Fig 3 misrepresents where the duplication event occurred – chANP32A duplication was downstream of SIM sequence, so “repeat 1” should have matching alignment of ch and hu. See Long 2015 Extended Data Fig 8 or Domingues 2017 Fig 2. Fig S3 alignment is inaccurate for the same reason. Figure S3 Legend indicates numbering with respect to huANP32A but this is not how the alignment is presented.

7. ANP32B and ANP32E were independently shown to be essential (in 1 of 5-7 screens each; http://ogee.medgenius.info/browse/). Please discuss the health of TKO cells.

8. Line 292 – these results are not “suprising” as they have already been reported for IAV:ANP3A interactions by others.

Reviewer #3: (No Response)

PLOS authors have the option to publish the peer review history of their article (what does this mean?). If published, this will include your full peer review and any attached files.

Reviewer #1: No

Reviewer #2: No

Reviewer #3: No

---

## [Decision Letter · Decision Letter 1]

12 Aug 2020

Dear Dr Wang,

Thank you very much for submitting your manuscript "Selective usage of ANP32 proteins by Influenza B Virus Polymerase: implications in determination of host range" for consideration at PLOS Pathogens. As with all papers reviewed by the journal, your manuscript was reviewed by members of the editorial board and by several independent reviewers. The reviewers appreciated the attention to an important topic. Based on the reviews, we are likely to accept this manuscript for publication, providing that you modify the manuscript according to the review recommendations. 

Sincerely,

Martin Schwemmle

Guest Editor

PLOS Pathogens

Ron Fouchier

Section Editor

PLOS Pathogens

Kasturi Haldar

Editor-in-Chief

PLOS Pathogens

orcid.org/0000-0001-5065-158X

Michael Malim

Editor-in-Chief

PLOS Pathogens

orcid.org/0000-0002-7699-2064

Reviewer Comments (if any, and for reference):

Reviewer's Responses to Questions

**Part I - Summary**

Reviewer #1: I am still very positive about this manuscript, and believe that it adds significantly to the field and our understanding of species-specific functional interactions between ANP32 proteins and influenza viruses. The data are novel, experiments well-executed, and the manuscript provides new detailed insights into the molecular basis of ANP32E usage, as well as avian versus mammalian ANP32A/B interplay with the polymerases of both influenza A and influenza B viruses. The findings have clear implications for trying to understand host restriction. Importantly, all of my previous comments, which were very minor, have been addressed satisfactorily.

Reviewer #2: The authors have addressed my major concerns. I appreciate their attention to each of the points raised and thorough responses.

Reviewer #4: The results presented in this manuscript advance our understanding of the important role of ANP32 proteins in the regulation and adaptation of the polymerase activity of Influenza A but most importantly on Influenza B viruses. It provides compelling and solid evidence, that ANP32 proteins promote IBV polymerase activity in human cells in a protein specific manner, which is in some parts divergent from the effects seen on the IAV polymerase.

Many of the primary research questions in this manuscript have been addressed before by Staller et al. 2019, Journal of Virology using similar methodology. Thus the originality and novelty of the subject is limited. In contrast to Staller et al. who reporter a certain degree of functional diversity between ANP32A and B proteins on the IBV Polymerase activity, the results in this manuscript show functional redundancy of both proteins in human cells. These differences may be attributed to the choice of IBV strain, cell type and method of generating KO clones and are in general interesting but not ground breaking.

The investigation of ANP32E, which shows limited support for the IBV polymerase is new and interesting, however, compared to ANP32A/B it appears to be of minor importance and its role in the species restriction of IBV to humans remains uncertain. The chosen methods mostly assess protein functions under non-natural (overexpression, purified) conditions and only very few assays verify results in the context of a viral infection, which overall is a weak point.

In the light of these shortcomings I would suggest submission of this manuscript to a more specialized journal.

In addition, I would like to pronounce my concerns about the response of the authors to reviewer #3, which is in parts rude and inappropriate. In a peer review process authors and reviewers should adopt a professional and supportive attitude even if the outcome is not as preferred.

To avoid any misinterpretation: My evaluation of the presented scientific work is by no means affected by this statement.

**Part II – Major Issues: Key Experiments Required for Acceptance**

Reviewer #1: None

Reviewer #2: (No Response)

Reviewer #4: (No Response)

**Part III – Minor Issues: Editorial and Data Presentation Modifications**

Reviewer #1: None

Reviewer #2: (No Response)

Reviewer #4: General remarks:

1. What is the rational of using a H7N9 IAV derived polymerase? This subtype has only recently jumped to humans but is not stably adapted (not circulating) to the human population in comparison to the H1N1 and H3N2 subtypes. Please include some information on why this strain was chosen as a comparison to IBV, which is a seasonal strain and much more stable adapted to humans.

2.The labeling of the y-axis in all figures displaying IAV polymerase activity using the polymerase reconstitution assay is wrong and needs to be changed to:

Firefly Luciferase activity instead of Firefly Polymerase.

Specific remarks:

1. lines 76-78: What means “the ANP32 family MAINLY includes ANP32A, ANP32B and ANP32E.”? Does this refer to in most species? Or expression levels? Are there other ANP32 proteins present? Please specify this statement.

2. lines 78: This sentence is misleading. Please acknowledge that also other groups have contributed to this finding.

3. Figure 1: Please indicate the subtype of the FluA polymerase as this is a relevant information for this assay.

4. Lane 123: what means: the last “functional member”? What function does this refer to? Please describe this in more detail. Also lane 124: what means “considered to be an important member of the ANP32 family”? Are there other non-important members? What is this hypothesis based on? Please provide more details.

5. Line 149: what means ANP32As? Is that s a typo? Please correct.

6. Figure 2: The title of this figure is misleading as it states that only ANP32As from different animals are investigated which is not true

7. Figure 2B: why is there no ANP32 protein band in the empty vector control lane? These are wild type cells and should express detectable levels of endogenous chicken ANP32 proteins. Please explain and interpret this result.

8. Figure 2C: Why were virus growth curves performed at 37°C? For IBV 33°C is commonly used as it also support much higher titers. I wonder whether overexpression of human ANP32B also has a positive effect at 33°C? Can you provide data on this?

9. lane 174: Please correct the sentence to …”that ARE all in short form…” and remove the are in lane 176.

10. Are there any other aa mutations between the different mammalian and avian type ANP32A proteins beside the 33 aa insertion? Please state so clearly in the primary text (in the respective section) as these mutations can also affect the protein function on the polymerase.

11. Which cells were used in figure 3d? Please also include this information in the text and legend.

12. lane 187: correct IVB to IBV

13. lane 204: I understand that chicken ANP32B also lacks the 33 aa insert. Is that correct? But there is a size shift in the westernblot bands compared to human ANP32B indicating addition amino acids. Please add this information to the beginning of this paragraph.

14. line 306: please change mutant to mutations

15. figure 7D: What happens if the LCAR region of ANP32A is removed? Is binding to IBV polymerase then abolished? Is that a common mechanism for all ANP32 proteins? Why was the ANP32A_165T mutant not tested in the co-IP assay but in the SPR?

16. line 360: I assume that the authors meant write morbidity instead of mobility? Please correct.

17: line 362: “polymerase replication” is a confusing description. Please substitute with “polymerase activity” Please substitute

18. overall the grammar, spelling and wording in the discussion part is of lower quality than the previous parts of the manuscript and needs major improvement (e.g. lines 417-425)

19. lines 420-425: The authors claim that the amount of mammalian-like ANP32A isoform in DF1 cells is not enough to support IBV replication. But in Figure 2C a growth kinetic of B/Yamagata/PJ/2018 is presented. So in principle, IBV replication seems to be supported in theses cells. Please rephrase this part of the discussion accordingly.

20. line 427: “chicken IAV”? does this mean avian IAV strains or do you really mean IAV from chicken? Please be precise.

PLOS authors have the option to publish the peer review history of their article (what does this mean?). If published, this will include your full peer review and any attached files.

Reviewer #1: No

Reviewer #2: No

Reviewer #4: No
---

## [Editor Report · Decision Letter 2]

17 Sep 2020

Dear Dr Wang,

We are pleased to inform you that your manuscript 'Selective usage of ANP32 proteins by Influenza B Virus Polymerase: implications in determination of host range' has been provisionally accepted for publication in PLOS Pathogens.

Best regards,

Martin Schwemmle

Guest Editor

PLOS Pathogens

Ron Fouchier

Section Editor

PLOS Pathogens

Kasturi Haldar

Editor-in-Chief

PLOS Pathogens

orcid.org/0000-0001-5065-158X

Michael Malim

Editor-in-Chief

PLOS Pathogens

orcid.org/0000-0002-7699-2064
---

## [Editor Report · Acceptance letter]

6 Oct 2020

Dear Dr Wang,

We are delighted to inform you that your manuscript, "Selective usage of ANP32 proteins by Influenza B Virus Polymerase: implications in determination of host range," has been formally accepted for publication in PLOS Pathogens.

Best regards,

Kasturi Haldar

Editor-in-Chief

PLOS Pathogens

orcid.org/0000-0001-5065-158X

Michael Malim

Editor-in-Chief

PLOS Pathogens

orcid.org/0000-0002-7699-2064